# MetaCaptioner: Towards Generalist Visual Captioning with Open-source Suites

**Zhenxin Lei**[4,1], **Zhangwei Gao**[2,1], **Changyao Tian**[6,1], **Erfei Cui**[2,1], **Guanzhou Chen** [2,1],
**Danni Yang**[3,1], **Yuchen Duan**[6,1], **Zhaokai Wang**[2,1], **Wenhao Li**[5,1], **Weiyun Wang** [3,1],
**Xiangyu Zhao**[2,1], **Jiayi Ji**[5], **Yu Qiao**[1], **Wenhai Wang**[6,1], **Gen Luo**[1]

[1]Shanghai AI Laboratory   [2]Shanghai Jiao Tong University   [3]Fudan University
[4]University of Chinese Academy of Science   [5]Xiamen University
[6]The Chinese University of Hong Kong

## Abstract

Generalist visual captioning goes beyond a simple appearance description task, but requires integrating a series of visual cues into a caption and handling various visual domains. In this task, current open-source models present a large performance gap with commercial ones, which limits various applications such as data synthesis. To bridge the gap, this paper proposes CapFlow, a novel multi-agent collaboration workflow. CapFlow demonstrates for the first time that, by capitalizing on open-source models, it is possible to achieve caption quality on par with GPT-4.1 in various domains with an 89.5% reduction in costs. By leveraging CapFlow as the data synthesizer, we produce high-quality visual captions from image and video domains at scale, and obtain a generalist visual captioner via fine-tuning, namely MetaCaptioner. Through extensive experiments, we show that MetaCaptioner not only achieves comparable captioning capabilities with commercial models but also reaches top-tier multimodal performance in the open-source community. We hope CapFlow and MetaCaptioner can benefit future multimodal research by providing a strong and cost-effective visual captioning solution. Our source code and models can be found at https://github.com/OpenGVLab/MetaCaptioner.

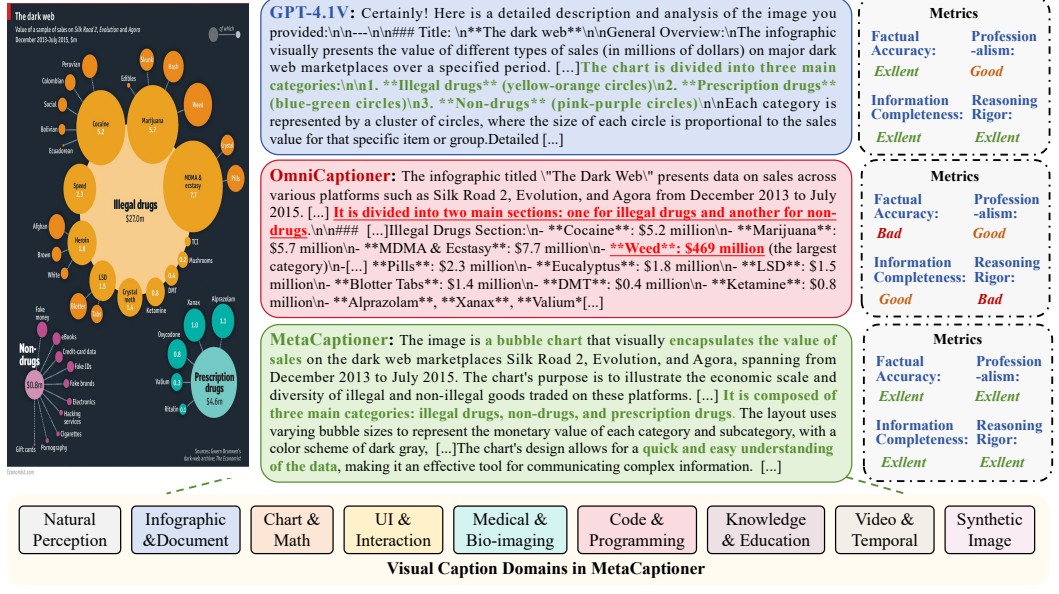

Figure 1: **Comparison of caption quality between GPT-4.1, OmniCaptioner (Lu et al., 2025) and MetaCaptioner.** These metrics are computed by GPT-5. MetaCaptioner is able to generate informative captions for a variety of visual domains with comparable quality to GPT-4.1.

# 1 INTRODUCTION

Visual captioning, a fundamental task in computer vision, aims to describe images or videos using natural sentences (Wang et al., 2020; Xu et al., 2015). Over the past decade, this field has received an influx of interest in the community (Chen et al., 2024b; Li et al., 2024d; You et al., 2016; Anderson et al., 2018), with the majority of research focusing on natural scene description (You et al., 2016; Yao et al., 2017; Anderson et al., 2018). With the advent of multimodal large language models (MLLMs) (Wang et al., 2025; 2024b; Li et al., 2024a), visual captioning has evolved beyond a simple multimodal task and now plays an increasingly crucial role in model pre-training (Wang et al., 2025; Betker et al., 2023) and data synthesis (Chen et al., 2024b; Li et al., 2024d; Yu et al., 2024). This progression imposes higher demands on existing visual captioners in handling more diverse and complex scenarios, such as mathematical figures (Zhang et al., 2024b; Lu et al., 2023b), diagrams (Mathew et al., 2022), and documents (Mathew et al., 2021).

To overcome this limitation, increasing efforts have been devoted to producing informative visual captions. Among them, a promising method is to design domain-specific prompt and leverage the advanced commercial MLLM, *e.g.,* GPT-4 (Hurst et al., 2024), to generate detailed captions (Chen et al., 2024b). While effective, such approach becomes prohibitively expensive as the data scale increases. Recent attempts also explore open-source MLLMs to synthesize the generalist visual captions (Xing et al., 2025; Lu et al., 2025), yet their caption quality still lags far behind that of commercial ones due to the limited capabilities. Therefore, one critical issues are naturally raised: *Is possible to bridge the gap in generalist visual captioning through open-source suites*?

To answer this question, we observe that the main difficulty of generalist visual captioning with open-source MLLM lies in the large domain gap. In particular, most MLLMs excel at natural scenes, and they tend to describe the subject, texture, and object relationship in an image. Nevertheless, the description requirements become quite different and challenging in other visual domains. For instance, for knowledge-based scenario, *e.g.,* math, the description should not only contain visual appearance, but also the related visual knowledge and underlying visual logic. As shown in Fig. 1, open-source MLLMs like OmniCaptioner (Lu et al., 2025) struggle to seamlessly integrate these capabilities into one description, often lacking fine-grained details in some aspects.

To address this issue, we propose *CapFlow*, a novel multi-agent collaboration workflow for producing generalist visual captioning with open-source MLLMs. The main principle of CapFlow is to decompose visual captioning into sub-tasks and synthesize evidence from diverse aspects. As shown in Fig. 2, CapFlow consists of a hierarchical workflow, where each agent performs as a distinct role in perception, visual knowledge extraction, visual reasoning, summary, *etc*. In CapFlow, visual descriptions from different perspectives are produced by the corresponding agents and ultimately aggregated into a single caption. To further eliminate the domain gap, we design a domain routing mechanism to dynamically assign the suitable workflow of CapFlow for different visual domains. Through close collaboration, CapFlow can achieve general visual captioning capabilities comparable to the top commercial models *i.e.* GPT-4.1 (OpenAI, 2025a), at only 10.5% of the cost.

Based on CapFlow, we aim to equip generalist visual captioning capabilities to common MLLMs (Wang et al., 2025), thus yielding an efficient and general visual captioner, namely *MetaCaptioner*. In particular, we leverage CapFlow as a strong data engine to produce informative captions for images and videos from various domains, with a strict reject sampling pipeline to filter out low-quality captions. Thanks to the cheap yet effective pipeline of CapFlow, we can efficiently scale up the data size to 4.1 millions, enabling a seamless convert of the open-source MLLM into MetaCaptioner through fine-tuning. Thanks to the high-quality captions from CapFlow, MetaCaptioner not only demonstrates powerful visual captioning capabilities at a lower cost ( 0.7% of GPT-4.1), but also achieves top-tier multimodal performance against existing open-source MLLMs.

To validate our approach, we conduct extensive experiments on 13 benchmarks of two common settings: (1) Visual reasoning with large language models (LLMs) (Lu et al., 2025): The detailed captions generated by CapFlow and MetaCaptioner are used as input to LLM for visual reasoning, thus diagnosing visual caption capabilities. (2) Downstream evaluations (Wang et al., 2025; 2024b; Xie et al., 2025): We directly evaluate the performance of MetaCaptioner on understanding and reasoning tasks across various domains, thereby validating the benefits of training with generalist caption data. Our experimental results show that our MetaCaptioner, even with a lightweight parameter size, can already achieve comparable captioning capabilities with the commercial model GPT-4.1 (OpenAI,

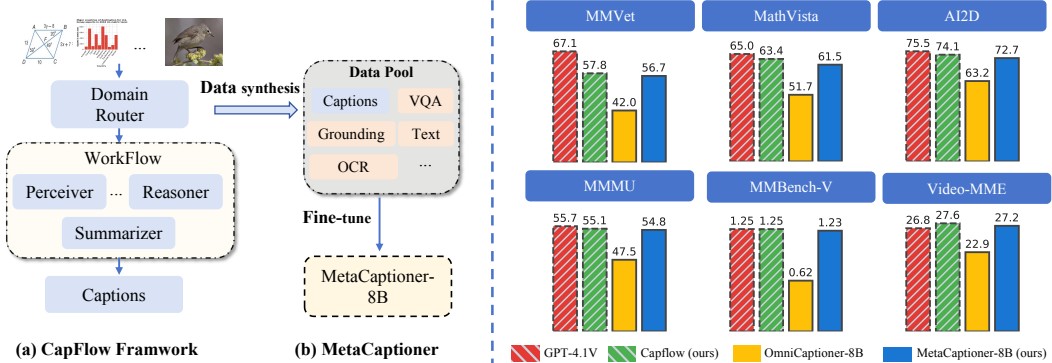

Figure 2: **Comparison of existing captioning systems with our our methods.** Our CapFlow adopts a multi-agent collaboration workflow to produce high-quality captions for training MetaCaptioner-8B. Under the setting of visual reasoning with LLMs, CapFlow and MetaCaptioner-8B achieve comparable performance with GPT-4.1 using open-source suites.

2025a) on several benchmarks, and its downstream performance also benefits from the synthetic data from CapFlow, *e.g.,* +3.7 on MathVerse. CapFlow and MetaCaptioner provide a cheap and strong visual captioning solution that will potentially benefit future multimodal research. In summary, our contributions are three folds:

- We propose CapFlow, an innovative multi-agent collaboration workflow for producing generalist captioning on various image and video domains. CapFlow is the first open-source framework that reaches comparable captioning performance with GPT-4.1.
- By leveraging CapFlow as the data synthesizer, we present the strongest open-source visual captioner in the community, namely MetaCaptioner. MetaCaptioner not only possess powerful generalist captioning capabilities, but also maintains top-tier multimodal performance.
- We conduct extensive experiments to ablate CapFlow and MetaCaptioner, providing invaluable hints for future research. In addition, our source models and codes will be publicly released, further advancing the development of generalist visual captioning and the broader multimodal research.

## 2 RELATED WORK

### 2.1 MULTIMODAL LARGE LANGUAGE MODELS

With the development of multi-modal large language models (MLLMs), visual-language alignment has attached more attention. Most approaches improve the alignment of visual and language by pre-training on large-scale image-text pairs (Radford et al., 2021; Alayrac et al., 2022; Li et al., 2022; 2023). Subsequent research attempt to achieve this challenge through incorporating high quality data for pre-training and supervised fine-tuning stage (Liu et al., 2023b; Li et al., 2024a; Guo et al., 2024b; Zhu et al., 2023; Wang et al., 2024b; Chen et al., 2024e; Wang et al., 2025). Some models further introduce more fine-grained image-text annotation data (Li et al., 2024d; 2025; Deng et al., 2025)(*e.g.* grounding and mask annotations) to reduce hallucinations and strengthen visual grounding ability. Nevertheless, due to the domain coverage and annotation costs, constructing multi-domain, high-quality image-caption pairs remains a significant challenge.

### 2.2 GENERALIST VISUAL CAPTIONING

Recent progress in multimodal community has significantly increased the demand for generalist visual captioning. In particular, existing methods can be broadly categorized into two categories. The first category often prompts powerful MLLMs (*e.g.*, GPT-4o (Hurst et al., 2024) and GPT-4V (Achiam et al., 2023) through carefully designed prompts to generate high-quality captions (Chen et al., 2024b;d; Lu et al., 2025). Some works also adopt coarse-to-fine-grained optimization like introducing visually differentiated descriptions (Chen et al., 2024d) and multi-turn caption optimization (Betker et al., 2023; Xing et al., 2025; You et al., 2025). The second category centers on multi-source caption synthesis, typically incorporating multiple domain expert models (*e.g.*, SAM (Kirillov et al., 2023),

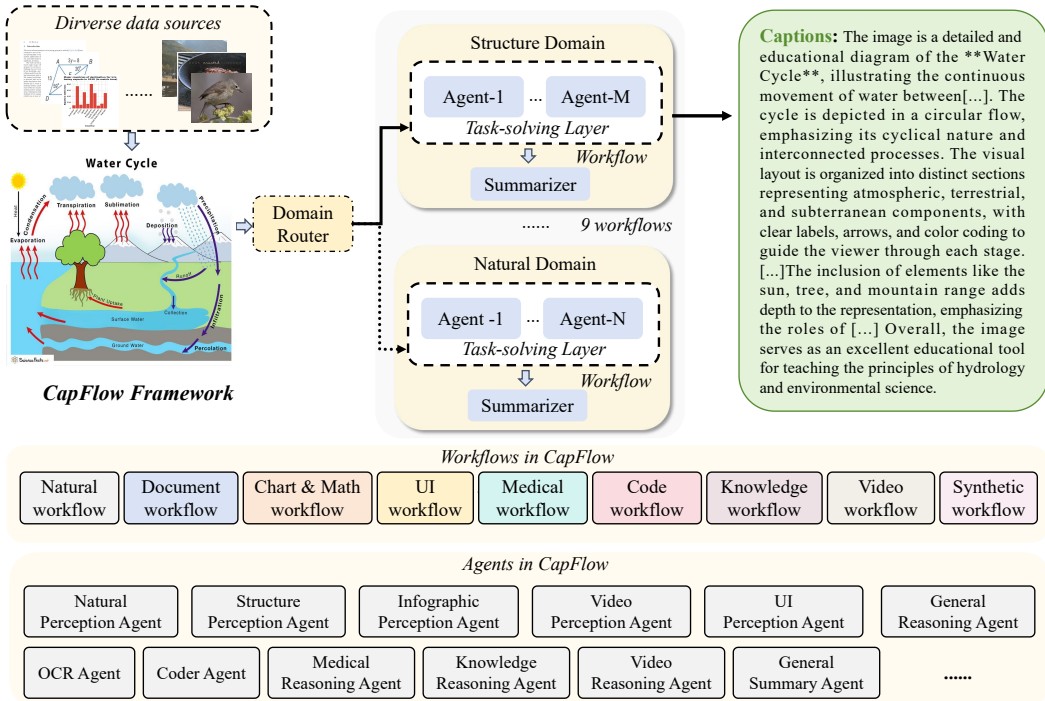

Figure 3: **Overview of CapFlow Framework.** CapFlow can dynamically select an appropriate caption workflow for the input image, and each workflow is equipped with multiple agents for close collaboration. With this pipeline, CapFlow can produce high-quality captions of various visual domains with open-source MLLMs, *e.g.,* Qwen2.5-VL (Bai et al., 2025).

Trace-Uni (Guo et al., 2024a), and Paddle OCR (Cui et al., 2025)) and merging the domain expert results (Li et al., 2024d; 2025; Wang et al., 2023a; Lian et al., 2025; Deng et al., 2025). However, both categories face limitations in content granularity and comprehension depth, often exceeding the capabilities of open-source MLLMs. Therefore, approaching generalist captioning with open-source models remains a significant challenge in the community.

# 3 CAPFLOW

## 3.1 OVERVIEW

A core challenge in generating diverse visual descriptions across different domains lies in the varying requirements for granularity and semantic depth (Xing et al., 2025; Lu et al., 2025). In this work, we propose CapFlow, a novel multi-agent collaboration framework for automated visual description. As illustrated in Fig. 3, CapFlow consists of two main parts: a *Domain Routing* and a *Hierarchical Captioning Workflow*. The design of CapFlow features in: 1) high-quality caption generation that integrates various visual cues, 2) collaborative agent framework applicable to a wide range of visual domains, and 3) cheap deployment cost and high scalability.

## 3.2 DOMAIN ROUTING

The motivation for the domain routing originates from an intuitive principle: *various visual domains demand varying levels of cognitive complexity* (Zhong et al., 2024; Chen et al., 2025). Therefore, CapFlow needs to first determine the domain of the visual input and then assign it a workflow suitable for visual captioning. To approach this target, we first adopt an MLLM as a domain router, with a carefully-designed prompt to query the visual domain of images. This process is similar to solving a visual question-answering problem, where the MLLM should select the best option from our pre-defined domain based on the visual input and the prompt. For data with a known domain, we can skip the domain routing part.

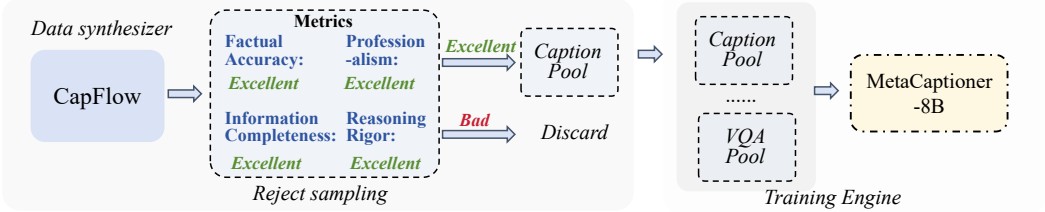

Figure 4: **Training pipeline of MetaCaptioner.** We leverage CapFLow as the data synthesizer to generate various captions, upon which a strict reject sampling pipeline is used to filter out low-quality ones. The obtained captions are combined with common instruction data to train MetaCaptioner.

## 3.3 Hierarchical Captioning Workflows

Based on the routing results, we design different routing paths for visual captioning of corresponding domains, and each routing path is customized for a visual domain. As shown in Fig. 3, the routing path is designed as a hierarchical workflow, with two distinct layers for task solving and information summarization, respectively.

In particular, the task-solving layer aims to decompose a complex captioning task into several simple sub-tasks and perform divide-and-conquer through multiple agents. Each domain is equipped with a distinct set of agents to solve domain-specific tasks, as shown in Tab. 1. Therefore, we design a variety of functional agents that are responsible for different tasks, where most of the agents are based on LLMs. In particular, there are a total of 30 functional agents in the workflow, which can be roughly divided into four categories: *guideline, perception, reasoning, and tools.* Specifically, the guideline agents aim to provide an overview description of the visual input, mainly containing global information like style, structure, and geography. In contrast, the perception agents depict the fine-grained details of images and videos such as textures, colors, and lighting. These two types of agents mainly focus on visual appearance. To compensate for high-level semantics, the reasoning agent is responsible for capturing the underlying visual knowledge, logic, and relationships. In addition, we notice that some domains often require additional visual tools to accommodate their requirements. For example, the document-related data often requires OCR parsers to obtain textual words. Therefore, we also design the tool agent to perform as specific tools like the OCR-based parser and the code parser. The detailed prompts are provided in the Appendix.

The visual cues generated by the task-solving layer are ultimately aggregated into one caption through the information summarization layer. In this layer, an LLM agent aims to summarize the textual contents from the task-solving layer one-by-one, analyze their contributions to the final captions, and organize into an informative, detailed and structured caption.

## 4 MetaCaptioner

Based on Capflow, we aim to train a strong generalist captioning model. To this end, we first collect images and videos from various domains, then employ Capflow as the data synthesizer to generate massive high-quality captions, and ultimately obtain MetaCaptioner-8B through fine-tuning.

### 4.1 Data Collection

**Data Source and Processing.** We first collect approximately 70 million raw images and videos from 140+ open-source image and video datasets. These sources span a wide range of domains, including natural images (*e.g.,* SAM (Kirillov et al., 2023) and LAION (Schuhmann et al., 2022)), structured images (*e.g.,* ArxivQA (Li et al., 2024c) and NovaChart (Hu et al., 2024)), knowledge-based images (*e.g.,* PMC-VQA (Zhang et al., 2023a) and iNat (Cui et al., 2018)), and aesthetic images (*e.g.,* PixArt (Chen et al., 2024a)). Due to the imbalance in different domains, we further collect a large mount of structured images (*e.g.,* charts, posters, GUI screenshots), text recognition images (*e.g.,* documents, reports), and commonsense images (*e.g.,* natural science, human activities). Following the protocol of (Zhu et al., 2025), we process these open-source datasets and obtain approximately 18 million structured images. To ensure the visual quality, we apply filtering based on image resolution

| Visual Domain | Functional Agent | Summary Agent |
|---|---|---|
| Natural | Natural Perception, General Reasoning, Visual Guideline | General Summary |
| Structure & Math | Structure Perception, Infographic Perception, General Reasoning, Visual Guideline | |
| Infographic & Document | Infographic Perception, OCR, General Reasoning, Visual Guideline | |
| Medical & Bio-Imaging | Natural Perception, Medical Reasoning, Visual Guideline | |
| UI & Interaction | UI Perception, OCR, General Reasoning | |
| Code & Programming | Coder, General Reasoning, Visual Guideline | |
| Knowledge & Education | Infographic Perception, Knowledge Reasoning, Visual Guideline | |
| Synthetic | Texture Perception, General Reasoning, Visual Guideline | |
| Video & Temporal | Video Perception, Video Reasoning, Video Guideline | Video Summary |

Table 1: **The agentic workflow of various visual domains.** We decompose the visual description task into visual guideline, perception, reasoning, and tool task.

and aspect ratio (Xing et al., 2025; Li et al., 2024d; 2025), retaining images with a short edge larger than 512 pixels, an aspect ratio less than 2, and videos with a resolution higher than 480p. As a result, we construct a high-quality dataset containing 5 million semantically rich and diverse samples from a broad range of sources. More details are provided in the Appendix.

**CapFlow as the Data Engine.** To accommodate data from diverse domains, we adopt CapFlow as the data engine for annotating the collected data to construct the dataset called MetaCaption-5M. For the functional agents, we adopt a powerful open-source MLLM Qwen2.5-VL-72B as the functional agent and carefully design task-specific prompts to guide the agent toward fine-grained visual perception and understanding. For the summary agent, we leverage the strong contextual processing capabilities of Qwen2.5-72B to integrate and summarize information from various functional agents, ultimately generating high-quality visual captions. The prompts for large-scale data synthesis are provided in the Appendix.

## 4.2 REJECT SAMPLING PIPELINE

As illustrated in Fig. 4, we further develop a strict reject sampling strategy to filter out low-quality data from the MetaCaption-5M dataset generated by CapFlow. Specifically, we employ carefully designed prompts to ask MLLMs to score the quality of each caption from multiple aspects. For the image modality, captions are evaluated across five dimensions: *Factual Accuracy*, *Information Completeness*, *Reasoning Rigor*, *Core Intent Capture*, and *Professionalism*. For the video modality, captions are assessed based on *Temporal and Factual Accuracy*, *Event and Detail Coverage*, *Temporal Causal Logic*, *Core Intent*, and *Professionalism*. To minimize the influence of the model's subjective judgment on the evaluation, we adopt a 3-point rating scale for scoring the results and only sample those that achieve professional levels (*e.g.*, rating 3) across all sub-domains for subsequent model training. As a result, we obtain a high-quality dataset called MetaCaption-4.1M.

## 4.3 TRAINING PIPELINE

Based on MetaCaption-4.1M, we trained a powerful multimodal large language model, namely MetaCaptioner-8B. Specifically, MetaCaptioner-8B adopts InternViT-600M (Chen et al., 2024e) and Qwen3-8B-Instruct (Yang et al., 2025) as the vision encoder and the language model, respectively. In the pre-training stage, we follow InternVL3.5 to perform native pre-training on various visual and text data (Wang et al., 2025). For instruction tuning, we mix our generalist captioning data (MetaCaption-4.1M) with instruction data from InternVL3.5 (Wang et al., 2025) to optimize the entire model. Training schedule and steps are kept consistent with InternVL3.5 (Wang et al., 2025).

## 5 EXPERIMENT

### 5.1 IMPLEMENTATION DETAILS

In CapFlow, we utilize Qwen2.5-VL-72B (Bai et al., 2025) and Qwen2.5-72B (Bai et al., 2025) as functional agents and summary agents, respectively. The entire annotation process requires approximately 480 H200 GPU days. For reject sampling, we employ Qwen2.5-VL-7B as the judger to provide the metric score, taking around 0.1 H200 GPU days. For pre-training of MetaCaptioner, it is trained for one epoch with a learning rate of 1e-5 and a batch size of 256. For supervised

| Model | MMMU | MMVet | Math Verse | Math Vista | Chart QA | Info VQA | AI2D | Video MME | Cost |
|---|---|---|---|---|---|---|---|---|---|
| GPT-4.1 | 55.7 | 61.7 | 56.8 | 65.0 | 62.3 | 63.2 | 75.5 | 26.8 | $1.47 |
| *Baseline* (7B) | 50.7 | 47.2 | 42.1 | 57.6 | 54.4 | 49.0 | 64.5 | 23.9 | **$0.01** |
| + Hierarchical Workflow | 51.6 | 48.6 | 44.0 | 59.0 | 57.8 | 44.2 | 66.0 | 26.1 | $0.02 |
| + Domain Routing | 54.7 | 50.5 | 43.9 | 58.7 | 58.0 | 45.0 | 67.4 | 26.2 | $0.02 |
| + Scale up to 72B | 55.1 | 57.8 | 53.1 | 62.5 | 59.2 | 50.2 | 74.2 | 27.6 | $0.14 |

Table 2: **Ablation study of CapFlow.** Our baseline is Qwen2.5-VL-7B (Bai et al., 2025). Through our careful design, CapFlow can achieve comparable performance to GPT-4.1 (OpenAI, 2025a) on most benchmarks, but at a much lower cost. We calculate the average inference cost on 100 samples.

fine-tuning (SFT), we train MetaCaptioner for 160k iterations with a learning rate of 2e-4. The complete training process takes approximately 192 H200 GPU days.

## 5.2 Evaluation Benchmarks

We evaluate our methods on 13 comprehensive multimodal benchmarks. Specifically, visual question answering benchmarks include InfographicVQA *test* (Mathew et al., 2022), ChartQA *test* (Masry et al., 2022), Document VQA (Mathew et al., 2021), and AI2D *test* (Kembhavi et al., 2016). MLLM benchmarks encompass MMBench *V1.1* (Liu et al., 2024), MMMU *val* (Yue et al., 2024), MMVet (Yu et al., 2023), MathVista *testmini* (Lu et al., 2023a), MathVerse *vision only* (Zhang et al., 2024b), SEEDBench2 Plus (Li et al., 2024b), and MMStar (Chen et al., 2024c). Video understanding benchmarks contain VideoMME (Fu et al., 2025) and MMBench-Video (Fang et al., 2024). The evaluation metrics follow existing methods using VLMEvalkit(Duan et al., 2024).

## 5.3 Evaluation Settings

**Caption Quality Evaluation.** Similar to the metrics in reject sampling, we define five metrics that can comprehensively reflect the caption quality: factual accuracy, completeness, reasoning rigor, intent capture, and professionalism. For these metrics, we design strict evaluation prompts for GPT-5 to assign a score from 1 (bad) to 3 (excellent) to each metric. Additionally, we conducted a thorough human evaluation to verify the alignment of different judges. Please refer to the Appendix for more details.

**Visual Reasoning with LLMs.** In this setting, visual captioners will generate description as the visual prompt for existing reasoning-enhanced LLMs, enabling them to seamlessly perform various multimodal downstream tasks. This setting decouples visual understanding from textual reasoning, where downstream performance highly depends on the generated captions. Therefore, we can quantitatively diagnose the informativeness of captions from the downstream performance.

**Direct Multimodal Evaluation.** Direct multimodal evaluation is the most common setting to assess the capabilities of MLLMs on downstream tasks (Wang et al., 2025; 2024b). Through this setting, we aim to diagnose the impact of high-quality synthetic data (*i.e., MetaCaption-4.1M*) on the multimodal capabilities of MetaCaptioner.

## 5.4 Ablation Study

### 5.4.1 Ablation Study of CapFlow

In Tab. 2, we ablate the design of CapFlow and compare it with GPT-4.1 (OpenAI, 2025a) under the setting of *visual reasoning with LLMs*. In this table, we first observe that starting from the Qwen2.5-VL-7B baseline, the introduction of the Hierarchical Workflow brings moderate gains on most benchmarks, *e.g.,* +1.4 on MMVet and +1.9 on MathVerse. The subsequent integration of the Domain Routing to categorize different domains yields more pronounced improvements, particularly on MMMU (+4.0 over baseline), MMVet (+3.3 gains), indicating its critical role in addressing domain gaps. Notably, scaling the base model of CapFlow to 72B parameters leads to substantial performance boosts across all tasks, nearing GPT-4.1 on several fronts: MMMU (55.1 vs. 55.7), MathVista (62.5 vs. 65.0), and AI2D (74.2 vs. 75.5). On VideoMME, CapFlow-72B even exceeds

| Model | Pre-training Data | SFT Data | MMMU | MMVet | Math Verse | Math Vista | Chart QA | Info VQA | AI2D | Video MME | Average |
|---|---|---|---|---|---|---|---|---|---|---|---|
| InternViT-600M + Qwen3-8B | ShareGPT4V-450K | Vanilla(3M) | 53.8 | 55.3 | 22.0 | 56.9 | 76.2 | 63.8 | 75.2 | 60.6 | 58.0 |
| | DenseFusion-450K | | 55.2 | 55.2 | **31.9** | 58.7 | 78.3 | 66.2 | 75.6 | 62.0 | 60.4 |
| | MetaCaption-450K | | **56.1** | **62.5** | 26.9 | **61.2** | **80.0** | **68.3** | **77.5** | **62.1** | **61.8** |
| InternViT-600M + Qwen3-8B | Vanilla(1M) | +ShareGPT4V-450K | 55.8 | 54.7 | 24.2 | 59.3 | 79.2 | 67.0 | 76.5 | 61.0 | 59.7 |
| | | +DenseFusion-450K | 55.2 | 62.1 | 27.1 | 59.0 | 78.4 | 68.3 | 77.2 | 60.7 | 61.0 |
| | | +MetaCaption-450K | **56.9** | **62.8** | 27.9 | 60.9 | 79.6 | **68.4** | 77.4 | 61.7 | **62.0** |

Table 3: **The impact of synthetic captions generated by CapFlow on pre-training and fine-tuning.** The vanilla data is randomly sampled from that of InternVL-3.5 (Wang et al., 2025). MetaCaptioner-450K is sampled from our synthetic data (MetaCaption-4.1M).

| Model | Factual Acc | Completeness | Reasoning Rigor | Intent Capture | Professionalism | Average |
|---|---|---|---|---|---|---|
| GPT-4.1 | **2.17** | **2.69** | **2.66** | **2.97** | 1.24 | **2.35** |
| CapFlow (ours) | 1.46 | 2.36 | 2.42 | 2.60 | **2.80** | 2.33 |
| Qwen2.5-VL-7B | 1.82 | 2.10 | 1.70 | 2.22 | 2.04 | 1.97 |
| OmniCaptioner | 1.27 | 1.93 | 1.43 | 2.33 | 1.17 | 1.63 |
| MetaCaptioner (ours) | **1.35** | **2.29** | **1.51** | **2.45** | **2.59** | **2.04** |

Table 4: **Caption quality of CapFlow and existing methods on 5 metrics.** We randomly sample 250 samples from multimodal benchmarks(Zhang et al., 2024b), and prompt GPT-5 to rate the five metrics.

GPT-4.1 (27.6 vs. 26.8), highlighting its strong capabilities in video domains. Most importantly, these results are achieved at only 10.5% of the cost of GPT-4.1 solutions, *i.e.,* $0.14 *vs.* $1.5 per image, making high-quality caption synthesis feasible for large-scale applications. In summary, the ablation study confirms that both the workflow design and model scaling are essential to the performance of generalist captioning.

### 5.4.2 Impact of Synthetic Captions on Training

In Tab. 3, we systematically evaluate the impact of integrating different types of synthetic caption data (Chen et al., 2024b; Li et al., 2024d) on pre-training and fine-tuning. As shown in Tab. 3, when augmenting pre-training data, CapFlow demonstrates clear advantages in several key benchmarks, with the highest scores on MMVet (62.5), InfoVQA (68.3), MMMU (56.1), and AI2D (77.5). Considering the very competitive baselines, these gains confirm the benefits of captions generated by CapFlow to MLLM pre-training. In the fine-tuning phase, incorporating CapFlow data also leads to notable improvements, particularly on MMVet (62.8) and MathVista (60.9), where it achieves the best average performance among all configurations. Notably, Video-MME results (61.7 for MetaCaption) highlight its competence in temporal visual reasoning, outperforming other synthetic data. In conclusion, the synthetic captions produced by CapFlow consistently enhance model performance across both training stages, demonstrating their richness, accuracy, and broad applicability.

### 5.5 Main Results

### 5.5.1 Comparison of Caption Quality

In Tab. 4, we compare the caption quality of our methods with several state-of-the-art captioning models on the complex multi-modal scene. Among open-source models, Qwen2.5-VL-7B (Bai et al., 2025) performs respectably (average score: 1.97), although it lags behind commercial models in factual and reasoning aspects. OmniCaptioner-7B (Lu et al., 2025) trails behind with an average of 1.63, indicating room for improvement in complex captioning scenarios. In contrast, our CapFlow framework achieves superior performance on reasoning rigor, intent capture, and professionalism with an average score of 2.33, slightly below the strong commercial baseline GPT-4.1 (2.35).

Our MetaCaptioner, fine-tuned with synthetic data from CapFlow, also demonstrates a much stronger generalist captioning ability, with performance comparable to GPT-4.1. These results confirm the strong captioning capabilities of both CapFlow and MetaCaptioner, effectively closing the performance gap with commercial models by using open-source tools.

| Captioner | LLM | MMB Video | Video MME | Math Vista | Math Verse | Math Vision | SEED2 Plus | Info VQA | MM Star | MMMU | MMB | AVG |
|---|---|---|---|---|---|---|---|---|---|---|---|---|
| Qwen2-VL-7B | DS-Qwen-7B | 0.57 | 20.8 | 47.7 | 40.5 | 31.6 | 56.6 | 46.0 | 43.8 | 42.4 | 54.7 | 37.4 |
| InternVL3.5-8B | DS-Qwen-7B | 0.88 | 26.2 | 60.6 | 44.7 | 34.8 | 61.8 | 48.6 | 52.7 | 52.8 | 55.4 | 38.7 |
| OmniCaptioner-7B | DS-Qwen-7B | 0.62 | 22.9 | 51.7 | 38.6 | 32.2 | 53.1 | 41.2 | 51.4 | 47.5 | 53.1 | 42.5 |
| MetaCaptioner-8B | DS-Qwen-7B | **1.23** | **27.2** | **61.5** | **47.8** | **37.2** | **62.7** | **49.0** | **53.3** | **54.8** | **57.8** | **49.4** |
| OmniCaptioner-7B | DS-Qwen-32B | 0.64 | 24.7 | 56.0 | 39.3 | 33.1 | 57.4 | 48.0 | 55.3 | 59.2 | 66.6 | 48.3 |
| MetaCaptioner-8B | DS-Qwen-32B | **1.49** | **26.7** | **65.1** | **49.9** | **38.5** | **66.5** | **57.0** | **57.5** | **66.8** | **74.4** | **55.3** |

Table 5: **Comparison of MetaCaptioner and existing captioners under the setting of visual reasoning with LLMs (Lu et al., 2025).** For each benchmark, we generate captions as visual prompts for LLM reasoning. We utilize Deepseek-R1-Distill-Qwen (Guo et al., 2025) as the LLM.

| Model | MMB Video | Video MME | Math Vista | Math Verse | Math Vision | Doc VQA | Chart QA | Info VQA | MM Star | MMMU | MMB | AVG |
|---|---|---|---|---|---|---|---|---|---|---|---|---|
| GLM4.1V-9B | 1.63 | 68.2 | 80.7 | 68.4 | 54.4 | 93.3 | 70.0 | 80.3 | 72.9 | 68.0 | 85.8 | 72.4 |
| Keye-VL-8B | - | 67.7 | 80.7 | 54.8 | 50.8 | 87.0 | 72.5 | 63.0 | 72.8 | 71.4 | 76.3 | - |
| Qwen2.5-VL-7B | **1.79** | 65.1 | 67.8 | 41.1 | 25.4 | **95.3** | **87.3** | **82.6** | 63.9 | 55.0 | **82.6** | 67.1 |
| MiniCPM-V2.6-8B | 1.70 | 60.9 | 73.3 | 35.0 | 21.7 | 90.8 | 82.4 | - | 57.5 | 50.9 | 78.0 | 60.9 |
| InternVL3-8B | 1.69 | **66.3** | 71.6 | 39.8 | 29.3 | 92.7 | 86.6 | 76.8 | **68.2** | 62.7 | 81.7 | 66.6 |
| InternVL3.5-8B-Instruct | 1.67 | 64.2 | 74.2 | 55.8 | 46.4 | 92.0 | 86.2 | 76.2 | 66.5 | 68.1 | 79.5 | 69.1 |
| **MetaCaptioner-8B** | 1.76 | 64.2 | **75.8** | **56.5** | **52.6** | 93.0 | 86.8 | 76.6 | 66.7 | **69.5** | 80.8 | **71.1** |

Table 6: **Direct performance comparison between MetaCaptioner and existing MLLMs.** For fair comparison, models with reinforcement learning are marked in gray.

### 5.5.2 RESULTS OF VISUAL REASONING WITH LLMs

In Tab. 5, we compare MetaCaptioner with existing captioners under the setting of visual reasoning with LLMs. In this setting, the LLM solves the multimodal task through the text prompt and captions produced by the captioner. When using Deepseek-R1-Distill-Qwen-7B (Guo et al., 2025) as the LLM, MetaCaptioner-8B outperforms both InternVL3.5-8B and OmniCaptioner-7B across all reported metrics. Notably, it achieves significant gains on MMBench-Video (1.23 vs 0.88 and 0.62), Video-MME (27.2 vs 26.2 and 22.9), and MathVision (37.2 vs 54.8 and 32.2). The average performance (AVG) of MetaCaptioner-8B reaches 49.4, substantially higher than OmniCaptioner-7B's 42.5. The performance gap further widens when using the larger LLM, *i.e.,* Deepseek-R1-Distill-Qwen-32B. MetaCaptioner-8B achieves significantly higher scores on challenging benchmarks such as Math Vista (65.1 vs 56.0), MMMU (66.8 vs 59.2), and SEEDBench2 Plus (66.5 vs 57.4). The significant performance gains indicate that MetaCaptioner generates more informative and structurally accurate captions that better leverage the enhanced reasoning capabilities of larger LLMs.

### 5.5.3 RESULTS OF DIRECT MULTIMODAL EVALUATION

In Tab. 6, we evaluate the general multimodal capability of MetaCaptioner and existing MLLMs (Yao et al., 2024; Hong et al., 2025; Team et al., 2025). From this table, MetaCaptioner demonstrates competitive performance across a wide range of multimodal benchmarks against leading open-source MLLMs. Notably, MetaCaptioner achieves strong results on several key benchmarks, including MathVerse (56.5), MathVision (52.6), and MMMU (69.5), outperforming most compared instruct models in these domains. Compared to models with reinforcement learning, MetaCaptioner also demonstrate better performance on some tasks, *e.g.,* +1.5 and +16.8 over GLM4.1V on MMMU and ChartQA, separately. This indicates its capability in handling knowledge-intensive and diagrammatic reasoning tasks, likely attributable to the high-quality caption data synthesized via CapFlow. These results affirm again that MetaCaptioner, trained with MetaCaption-4.1M, achieves top-tier multimodal capabilities among open-source models.

## 6 CONCLUSION

In this work, we introduce CapFlow, a novel multi-agent collaboration framework designed to bridge the performance gap between open-source and commercial generalist visual captioning systems. By efficiently orchestrating multiple specialized agents, CapFlow achieves comparable caption quality with GPT-4.1 across diverse visual domains with an 89.5% reduction in costs. Leveraging this

scalable pipeline, we synthesize a large-scale high-quality caption dataset spanning both image and video modalities, which in turn enable the training of our generalist captioner, namely MetaCaptioner. Extensive experiments confirm the strong generalist captioning ability and multimodal capabilities of MetaCaptioner against existing MLLMs. We believe that CapFlow and MetaCaptioner will serve as valuable tools for future research and applications in visual captioning and reasoning and the broader multimodal community.

## 7 Ethic Statement

This study exclusively utilizes images and videos from open-source datasets, which have been rigorously filtered and screened to mitigate potential biases and ethical issues. Furthermore, all pre-trained models employed are sourced from the open-source community. However, despite these precautions, large models are inherently susceptible to hallucinations, which may lead to the generation of misleading or incorrect content.

## 8 Reproducibility Statement

For reproducibility, our paper provides sufficient implementation details, hyperparameters and a complete set of prompts in the main text and the Appendix. To further facilitate replication, all source codes and models are now available at https://github.com/OpenGVLab/MetaCaptioner.

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

| Parameters | Pre-training Stage | SFT Stage |
|---|---|---|
| batch size | 256 | 256 |
| iterations | 300, 000 | 160, 000 |
| token length | 32, 768 packed | 32, 768 packed |
| optimizer | AdamW | AdamW |
| learning rate | 1e-5 | 2e-5 |
| training module | InternViT-300M + MLP Projecter + Qwen3-8B | InternViT-300M + MLP Projecter + Qwen3-8B |

Table 7: **Training details for MetaCaptioner-8B.**

# A  APPENDIX

## A.1  THE USE OF LARGE LANGUAGE MODELS (LLMs)

We only use the LLM (Deepseek-R1) to polish papers, but not for idea generation or to participate in the actual writing process.

## A.2  TRAINING DETAILS

We have trained the Qwen-3-8B-Instruct model for two days on 128 H200 GPUs, resulting in the MetaCaptioner-8B model. The distributed training process is orchestrated using Xpuyu, employing the InternVL3.5 (Wang et al., 2025) optimization strategy. Additional hyper-parameters utilized during training are detailed in Tab. 7.

## A.3  ANALYSIS ON METACAPTION DATASET

In this section, we provide a comprehensive overview of our dataset, including the data sources, domain partitioning methodology, and a thorough statistical analysis.

### A.3.1  DATA SOURCE

As illustrated in Fig. 5, the MetaCaptioner-4.1M dataset is derived from over 140 open-source datasets. Data acquisition is conducted through a combination of automated collection and manual sampling, with the entire process spanning one year. The specific sources for these datasets are enumerated as follows:

**Natural.** Within the domain of natural images, we primarily focus on urban landscapes, natural scenery, human activities, animals, plants, still life, dynamic objects, and remote sensing images. Our data sources in these categories include: SAM (Kirillov et al., 2023), COCO (Lin et al., 2014), All Seeing project (Wang et al., 2023b; 2024c), Laion (Schuhmann et al., 2022), Crowdhuman (Shao et al., 2018), Inifinity-mm (Gu et al., 2024), CityScapes (Cordts et al., 2016), etc.

**Structure & Math.** Within the domain of structured and mathematical images, our focus encompasses charts, tables, diagrams, equations, and geometric figures, all of which are characterized by clear mathematical and numerical information. We collect raw data from the following open-source datasets, *e.g.*, Pixmo (Deitke et al., 2024), IM-TQA (Zheng et al., 2023), MMC (Liu et al., 2023a), Geomverse (Kazemi et al., 2023), KVQA (Sanket Shah & Talukdar, 2019), etc.

**Infographic & Document.** We gather infographic and document-related data from the sources listed below: InfoVQA(Mathew et al., 2022), Pixmo(Deitke et al., 2024), ArxivVQA(Li et al., 2024c), Laion(Schuhmann et al., 2022), etc. Compared with the Structure & Math domain, this domain is assigned specifically for those images with large amount of text information.

**Medical & Bio-Imaging.** For medical and biology images, which require specialized domain knowledge for accurate interpretation, we collect and augment data from open-source medical datasets. The specific data sources are as follows: Red-VQA (Gai et al., 2025), ImageClef (Zhou et al., 2019), PMC-VQA (Zhang et al., 2023a), and Path-VQA (He et al., 2020), etc.

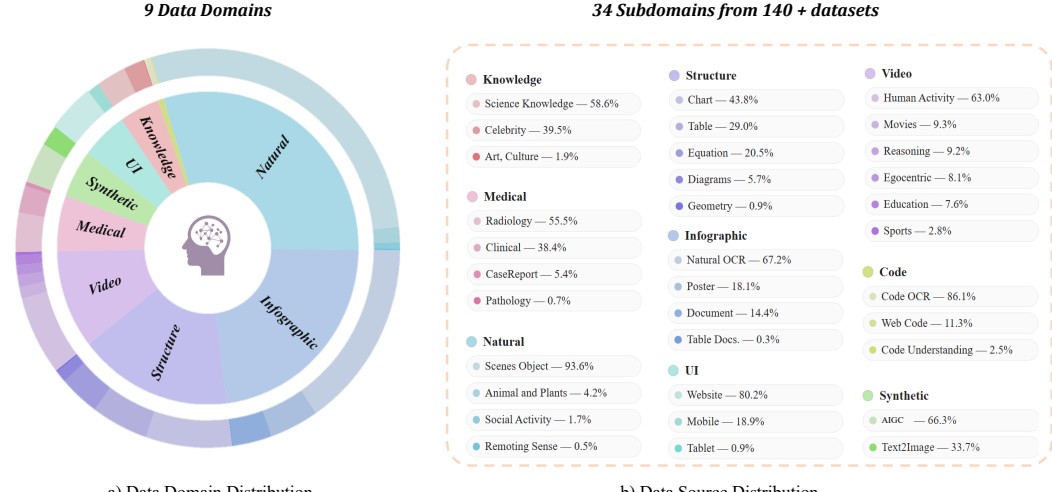

*9 Data Domains* — *34 Subdomains from 140 + datasets*

a) Data Domain Distribution — b) Data Source Distribution

Figure 5: **Category distribution of MetaCaption-4.1M.** a) The distribution of 9 main domain; b) The data distribution in each subdomain. We report the percentage of each subdomain of its corresponding main domain.

**UI & Interaction.** In the UI & Interaction domain, we collect web page screenshots, mobile screen captures, and other images related to web user interfaces. The following datasets are utilized for further data collection: Taperception (Schoop et al., 2022), Amex (Chai et al., 2024), Android in the zoo (Zhang et al., 2024a), UIBert (Bai et al., 2021), etc.

**Code & Programming.** In this domain, we collect screen snapshots, augmented code through transformations, as well as structured code diagrams. The following datasets are utilized for further data collection: MMMU (Yue et al., 2024), WebSight (Bhathal & Gupta, 2025), and Verillog (Pinckney et al., 2024), etc.

**Knowledge & Education.** To address the need for comprehensive world knowledge, we construct a knowledge dataset divided into social science and natural science domains. The dataset is sampled from a variety of sources, including high school-level examinations, natural science questions, cultural festivals, and real-world social scenarios, *e.g.*, ScienceQA (Saikh et al., 2022), ImageNet (Deng et al., 2009), Inat (Cui et al., 2018), GAOKAOBench (Zhang et al., 2023b), MMMU (Yue et al., 2024), etc.

**Synthetic & Asthetic.** For synthetic and aesthetic images, we collect data from open-source AIGC datasets, such as LAION, PixArt, and openly licensed photography websites. Specifically, we collect data from the followin datasets, *e.g.*, PixArt-$\sigma$ (Chen et al., 2024a), COCO-T2I (Lin et al., 2014), and Laion-T2I (Schuhmann et al., 2022), etc.

**Video & Temporal.** We collect temporal data with semantic-rich information from human activities, educational videos, movies, egocentric videos, reasoning videos, and sports videos, *e.g.*, LSMDC (Rohrbach et al., 2017), ShareGemini (Share, 2024), ShareGPTVideo (Chen et al., 2024d), VideoChatGPT (Maaz et al., 2023), SSV2 (Goyal et al., 2017), Clevrer (Yi et al., 2019), Egotask (Jia et al., 2022) etc.

Finally, we meticulously select high-quality samples from each dataset, and, following the protocols of (Zhu et al., 2025), perform additional filtering, augmentation, and deduplication to ensure data quality.

### A.3.2 DOMAIN SPLIT

As summarized in Tab. 8, the MetaCaptioner dataset encompasses nine principal categories: Natural, Structure & Math, Infographic & Document, Medical & Bio-Imaging, UI & Interaction, Code & Programming, Knowledge & Education, Synthetic & Aesthetic, and Video & Temporal. Each major

| Domain | Subdomain | Avg. Caption Length |
|---|---|---|
| Natural | Scenes Object, Social Activity, Animal and Plants, Remoting Sense, | 863 |
| Structure & Math | Chart, Table, Equation, Geometry, Diagram | 868 |
| Infographic & Document | Natural OCR, Document, Poster, Table Docs. | 836 |
| Medical & Bio-Imaging | Radiology, Pathology, Clinical, Case Report | 729 |
| UI & Interaction | Website, Mobile, Tablet | 1033 |
| Code & Programming | Code OCR, Code Understanding, Web Code | 1150 |
| Knowledge & Education | Science Knowledge, Art & Culture, Celebrity | 916 |
| Synthetic & Aesthetic | Text2Image, AIGC | 679 |
| Video & Temporal | Human Activity, Education, Movies, Egocentric, Reasoning, Sports | 904 |

Table 8: **Domain split and static results for MetaCaption-4.1M.** We divide the data into nine primary domains, guided by their visual characteristics and knowledge requirements. The subdivision into specific subdomains is primarily based on the data sources and their original tasks. We further static the average caption length for each domain.

| Model | Factual Acc | Completeness | Reasoning Rigor | Intent Capture | Professionalism | Human Preference | Average |
|---|---|---|---|---|---|---|---|
| GPT-4.1 | 2.80 | 2.66 | 2.88 | 2.82 | 2.65 | 2.89 | 2.76 |
| **CapFlow (ours)** | **2.82** | **2.95** | **2.93** | **2.85** | **2.95** | **2.96** | **2.90** |
| Qwen2.5-VL-7B | 2.41 | 2.32 | 1.97 | 2.54 | 2.33 | 1.84 | 2.31 |
| OmniCaptioner-7B | 2.42 | 2.37 | 2.53 | 2.51 | 2.44 | 2.33 | 2.46 |
| **MetaCaptioner (ours)** | **2.88** | **2.94** | **2.86** | **2.90** | **2.90** | **2.90** | **2.89** |

Table 9: **Caption quality of CapFlow and existing methods on 5 metrics with human evaluation.** We randomly sample 250 samples from 5 multimodal benchmarks (Yue et al., 2024; Wang et al., 2024a; Kembhavi et al., 2016; Mathew et al., 2022; Chen et al., 2024c) and invite eight human annotators to evaluate the captions according to five metrics.

category is further subdivided into 34 subdomains based on the visual content and thematic focus. The corresponding visual understanding tasks include classification, detection, segmentation, visual grounding, common VQA, image captioning, multi-round conversation, OCR, document OCR, chart QA, chart OCR, table OCR, table QA, mathematical problem solving, visual reasoning, diagram reasoning, medical VQA, knowledge VQA, video captioning, video VQA, video highlight detection, video conversation, among others.

## A.4 HUMAN EVALUATION OF CAPTION QUALITY

We conduct an additional experiment to further evaluate caption quality based on human judgment (Zhang et al., 2025; Chen et al., 2024b). Specifically, we select 250 data samples, as illustrated in Tab. 4, to construct the evaluation dataset. Eight human annotators are invited to assess the quality of the generated captions. The assessment criteria include: factual accuracy, completeness, reasoning rigor, intent capture, professionalism, and a human preference score. Annotators are instructed to follow a rigorous three-point rating protocol for all criteria (Fang et al., 2024). During the evaluation, we adhere to a strict double-blind protocol: each annotator is randomly assigned captions generated by Qwen2.5-VL-7B, OmniCaptioner, MetaCaptioner, Capflow-72B, and GPT-4.1, with model identities concealed.

The results, summarized in Tab. 9, demonstrate strong consistency with the findings in Tab. 4, thereby validating the effectiveness of our reject sampling strategy and mitigating the potential model bias introduced by LLM-as-judge. Moreover, we find that human annotators assigned higher reasoning rigor scores to the fine-grained captions produced by MetaCaptioner and CapFlow, compared to those generated by GPT-4.1 and OmniCaptioner. We argue that detailed captions with clear visual cues and explicit reasoning processes are more favored by humans.

## A.5 ANALYSIS ON THE ROBUSTNESS OF DOMAIN ROUTER

We conducted an ablation study to assess the robustness of the domain router. Specifically, four human annotators carefully selected 500 images from the original training data, covering eight major visual domains. These images feature complex scenes with mixed-domain characteristics, such as geometric figures accompanied by surrounding text and equations, or posters that combine natural scenes with overlaid captions. To ensure annotation quality, blind labeling and cross-validation were

| Visual Domain | Accuracy |
|---|---|
| Human Average | 97.0% |
| Average | 84.8% |
| Natural | 85.7% |
| Structure & Math | 87.5% |
| Infographic & Document | 83.3% |
| Medical & Bio-Imaging | 100.0 % |
| UI & Interaction | 77.2% |
| Code & Programming | 83.2% |
| Knowledge & Education | 82.9% |
| Synthetic | 80.0% |

Table 10: **Robustness analysis on domain routing mechanism.** We carefully selected 500 samples from the MetaCaptioner-4.1M dataset and invited four human annotators to conduct standard ground truth annotation and subsequent evaluation.

| Model | LLMs | MMMU | MathVista | AI2D | Overall | Cost |
|---|---|---|---|---|---|---|
| GPT-5 | DS-Qwen-7B | **58.2** | 64.0 | 74.2 | 65.4 | 1.56$ |
| Gemini-2.5-pro | DS-Qwen-7B | 56.8 | 63.8 | **76.2** | **65.6** | 2.33$ |
| GPT-4.1 | DS-Qwen-7B | 55.7 | **65.0** | 75.5 | 65.4 | 1.47$ |
| Gemini-2.0-Flash | DS-Qwen-7B | 56.0 | 56.6 | 74.0 | 62.2 | 0.37$ |
| Capflow-72B | DS-Qwen-7B | 55.1 | 62.5 | 74.2 | 64.0 | **0.14**$ |

Table 11: **Caption quality comparison between Capflow-72B and the latest strong commercial models.** Capflow-72B demonstrates a comparable result with the latest commercial models. We calculate the average inference cost on 100 samples.

performed, with consensus required to resolve any disagreements. Following the captioning stage, all four annotators evaluated both the routing and captioning results for each misrouted sample to further assess any performance degradation. The detailed evaluation results for each visual domain are provided in Tab. 10.

As shown in Tab. 10, we observed 76 misrouted samples in our experiments, which were subsequently subjected to additional human evaluation. Despite routing errors, the captions generally maintained a high professional standard: four annotators rated 64 out of the 76 misrouted samples as acceptable (84.2%). During the final quality filtering stage, 83.3% of misrouted samples exhibiting clear hallucinations or errors were accurately excluded.

Our findings suggest that, in most cases, misrouting leads only to sub-optimal outputs rather than significant quality degradation. Only a small proportion of misrouted samples adversely affect caption quality, and these can be effectively filtered out during the reject sampling stage.

For mixed-domain inputs, the domain router assigns workflows based on the most prominent visual features. In general, Qwen2.5-VL-7B demonstrates strong performance in this task. However, our analysis of misclassified cases indicates that rich-text content and numerical tables can substantially interfere with the model, resulting in misclassification among the Knowledge, Document, and Structure Image domains. Furthermore, AIGC-generated images and natural scenes can also introduce confusion in certain real-world scenarios.

A.6    COMPARISON WITH COMMERCIAL BASELINES

We conduct a thorough comparison between Capflow-72B, Gemini-2.0-Flash, Gemini-2.5-Pro (Comanici et al., 2025), GPT-4.1 (OpenAI, 2025a), and GPT-5 (OpenAI, 2025b) in the setting of visual reasoning with LLMs. This setting is designed to evaluate the quality of the caption in an LLM-driven manner. Specifically, the whole evaluation process decouples the visual understanding into

two stages: VLLM perception and LLM reasoning. First, the caption model is prompted to generate the caption for images, and then the LLM will use the caption to answer the question directly. Thus, a higher score in this setting can effectively reflect the completeness, accuracy, and visual consistency of the caption.

As shown in Tab. 11, Capflow-72B outperforms Gemini-2.0-Flash (64.0 vs. 62.2) and achieves performance comparable to GPT-5 (64.0 vs. 65.6). We observe that Capflow demonstrates a notable advantage in the domains of mathematical and structured images. For example, on the MathVista benchmark, Capflow, Gemini-2.0-Flash, and GPT-5 achieve scores of 62.5, 56.6, and 64.5, respectively. On the AI2D benchmark, Capflow even matches the performance of GPT-5 (74.2 vs. 74.2), underscoring its strong capability in structured image captioning. Although there remains a performance gap between Capflow and the strongest commercial baseline, Gemini-2.5-Pro and GPT-5 (64.0 vs. 65.5 vs. 65.4), Capflow-72B exhibits significantly lower inference costs (0.02$ vs. 2.33$ vs. 1.56$ for 100 samples).

### A.7 PROMPT TEMPLATES

In this section, we first present the prompt templates used in Capflow for constructing functional agents and summary agents. Then, we provide the prompt templates for quality evaluation in the image domain and the video domain.

#### A.7.1 PROMPT TEMPLATES USED IN CAPFLOW

The prompt templates used in Capflow contain three parts: domain router agent, functional agent, and summary agent. We will detail these prompt templates in the following.

**Domain Router Agents.** The domain router agents in Capflow are designed to assign workflows based on the primary characteristics of the visual input. To mitigate potential misclassification risks, we explicitly define the distinguishing features of each visual domain in the prompt design and provide clear boundaries for features that are prone to confusion. Furthermore, we introduce a three-point confidence assessment mechanism to further reduce routing errors. The agent is required to evaluate its own routing confidence, assigning lower confidence scores to mixed-domain or ambiguous cases. With these two explicit guidance strategies, the domain router becomes capable of efficiently allocating workflows across large-scale, multi-source visual inputs. The detailed prompt template utilized to activate domain router agent is presented as follows:

---

**Prompt for Domain Router Agent**

**You are a rigorous visual domain classification agent. Your task is to classify the input image or video (optionally with user-provided text) into one of the following predefined categories and return a class-JSON object with the following structure:**

**Output requirements:**

- The output must be a single-line class-JSON object. The keys and their order must be strictly: {class, explanation, confidence_score}.
- `class`: One of the predefined English category strings below.
- `explanation`: A brief, objective analysis (1-3 sentences), indicating key visual features and reasons for not choosing similar categories.
- `confidence_score`: An integer from {1, 2, 3}, following the scoring criteria below.
- No extra fields, markup, line breaks, lists, apologies, or disclaimers are allowed.

**Predefined Categories (exact strings):**

- Natural
- Structure & Math
- Infographic & Document
- Medical & Bio-Imaging
- UI & Interaction
- Code & Programming
- Knowledge & Education
- Synthetic & Aesthetic
- Video & Temporal

**Category definitions and classification hints:**

- **Natural**: Real-world scenes, human social activities, plants/animals in nature, natural landscapes, panoramas, satellite/drone imagery.
- **Structure & Math**: Mathematical charts, tabular images, geometric figures, formulas, mathematical flowcharts, or diagrams emphasizing quantitative or formal relationships. The core feature is clear representation of mathematical/scientific content.
- **Infographic & Document**: Natural scenes with prominent text, document pages (e.g., PDFs), posters with large blocks or prominent titles of text, focusing on layout understanding and text extraction.
- **Medical & Bio-Imaging**: Radiology (CT/MRI/X-ray), pathology slides, clinical photos, multi-image case reports, images featuring pathological traits in humans, animals, or plants.
- **UI & Interaction**: Screenshots of websites/mobile/tablet/app interfaces, web pages, or interactive UI, characterized by app icons or clear web/UI elements.
- **Code & Programming**: Screenshots of code, code analysis, IDE/editor terminals, code fragments—features clear code content covering more than 75% of the image.
- **Knowledge & Education**: Images with biological, physical, chemical, or other scientific diagrams, educational illustrations with text, scientific visualizations, art/culture, museum exhibits relevant for learning or general knowledge.
- **Synthetic & Aesthetic**: All AI-generated images, typically with stylized features such as anime, AI-synthesized humans, etc.

---

- **Video & Temporal**: Content emphasizing temporal dynamics, such as human activities, instructional videos, film clips, first-person views, motion analysis, or temporal reasoning.

**Disambiguation rules:**

- **Video frames**: If the core task is clearly about time/motion/behavior recognition, choose "Video & Temporal"; otherwise, classify by content visible in the frame.
- **Charts in documents**: If layout dominates, choose "Infographic & Document"; if the chart contains rich quantitative data, choose "Structure & Math".
- **IDE code interfaces**: Choose "Code & Programming" rather than "UI & Interaction".
- **Educational posters/handouts**: If containing clear biology, physics, chemistry, culture, or history knowledge, choose "Knowledge & Education" rather than "Infographic & Document".
- **Natural scenes**: If obviously stylized, consider "Synthetic & Aesthetic" instead of "Natural".
- **Mixed-domain images**: Choose the most appropriate category based on the dominant content, and explain the reasoning (e.g., a table mixing numbers and text, if inferable by numbers, choose "Structure & Math").

**Confidence scoring (strict 3-point scale):**

- 3 = Clear, unambiguous match; multiple strong features indicate a single category; low overlap with others.
- 2 = Reasonable match but some ambiguity or overlap; features present but not complete (e.g., large text blocks—Document or Math?).
- 1 = Low confidence; poor image quality, heavy occlusion, mixed domains with no dominant type, or missing key features.

**Functional Agents.** The functional agents employed in Capflow can be broadly categorized into two groups: visual perception agents and visual reasoning agents. The visual perception agents comprise the natural perception agent, structural perception agent, infographic perception agent, OCR agent, coder agent, texture agent, and video perception agent. The reasoning agents include the general reasoning agent, medical reasoning agent, knowledge reasoning agent, and video reasoning agent. The detailed prompt templates utilized to activate each functional agent are presented as follows:

---

### Prompt for Natural Perception Agent

**You are an expert specializing in ultra-high-detail image description and style interpretation. Your task is to craft exceptionally detailed, logically clear, and fluent descriptions for every image, identifying its artistic medium, decorative features, and rendering techniques, to serve as figure captions for professional academic publications.**

You should describe a variety of image types, including but not limited to classical and contemporary artworks, classical musical scores, science posters, web interfaces, natural landscapes, close-ups of plants and animals, human activities, sports, street scenes, ancient and modern architecture, literary works, economic statements, mathematical charts, chemical formulas, physical model diagrams, film posters, and other images rich in information.

**Please follow these rules when describing:**

1. **For all concrete objects and artistic representations** (*e.g.*, road signs, pedestrians, animals, cups, painted figures, etc.): Accurately identify all visual subjects in the image and provide a precise and comprehensive description of their category, key features, color (type, brightness, saturation), geometric characteristics (square, round, irregular shapes), quantity, size, local details (such as the jacket and pants of a person, accessories on clothing, etc.), texture features, absolute and relative position in the image, and spatial relationships (such as objects stacked or laid flat).

2. **For visual elements with abstract referential meaning or requiring domain-specific knowledge** (*e.g.*, charts, design elements in posters, stacks of geometric bodies in space, quadratic function graphs, etc.): Use domain-appropriate vocabulary and sentences to accurately and thoroughly describe their type, key features, geometric shape, quantity, size, local details (such as accessories on clothing, inflection points of a quadratic function, points, lines, planes in geometric images, circular borders, etc.), absolute position within the overall image, relative position to other objects, and all decorative features (such as icon outlines).

3. **For all visible text elements:** Accurately and completely identify all visible text and its features, presenting them in a readable form, including text and numbers in documents, formulas, tables, charts, and artistic typography. Additionally, accurately identify the features of these texts (such as font, color, size, layout characteristics), and separately describe any distinctive text features, including bold, underlined, special fonts, different colors, or highlighted text.

4. **For image backgrounds:** Accurately indicate the arrangement of environmental elements in concrete scenes as well as the dynamic and atmospheric characteristics those scenes express (*e.g.*, a soccer field during a match, a clean and tidy classroom, or a rapidly flowing waterfall). For abstract or artistically colored image backgrounds, provide an overall summary (*e.g.*, a pure white background, a poster with a hand-drawn style globe).

5. **Describe the style features of all visible objects** (including but not limited to lighting, brightness and darkness, color tone, realism, contrast, saturation, etc.), and for images with significant aesthetic characteristics, moderately integrate the atmosphere created by the image and the possible sensory experience (such as light and shadow, texture, ambience, and a sense of movement). All stylistic features should be naturally incorporated into the descriptions of visual elements and the overall background, and must not be listed as independent tags.

6. **If there are special logical relationships in the image** (such as spatial arrangement, inclusion, nesting, flow, causality, etc.), describe these in order according to their logical characteristics.

7. **For obscured, blurred, or extremely small elements:** Only describe the identifiable portions; parts that cannot be accurately recognized can be omitted.

8. **All visual and textual elements in the natural and social sciences** must be described using professional terminology; for artistic, aesthetic, and literary content, you are encouraged to use elevated adjectives and expressive vocabulary to convey the emotional impact and aesthetic qualities of the image.

9. **If common knowledge or world knowledge is involved** (such as plants and animals, celebrities, famous landmarks, historical sites, etc.), you must state it explicitly and avoid vague expressions like "someone" or "somewhere."

10. **Output should be in complete, coherent paragraphs**, allowing multiple sentences per paragraph, and avoiding lists, structured subheadings, bullet points, or semantic tags.

11. **When describing objects, data, or structures**, you are encouraged to use measurable language or relative references (such as "in the lower left corner of the frame," "as tall as the figure," "in sharp contrast with the background").

12. **Describe the content of the image directly**, without using phrases like "the image shows" to start. Avoid repetition and uncertain expressions, do not use speculative language such as "possibly" or "about," do not describe content not present in the image, and do not use phrases like "there is no text in the image."

13. **DO NOT infer or analyze any content in the image;** only describe what is visibly present.

14. **For complex images**, keep the total description within 500 words, with priority given to a complete presentation of the main structures and prominent features; details may be appropriately condensed.

---

## Prompt for Logical Perception Agent

**You are an expert with outstanding professional competence in spatial understanding, logical reasoning, and mathematical analysis. Your task is to produce a highly information-dense, detail-rich, and rigorous academic description in the style of a research paper for images that require professional analysis and logical reasoning in the natural sciences and interdisciplinary fields, including but not limited to mathematics, physics, chemistry, electronic information, economics, psychology, pharmacy, and medicine. Your description will serve as the figure caption for images in professional academic publications.**

The description should adhere to academic conventions, employ strict logical structure, precise terminology, and fluent written expression, appropriately embed LaTeX formulas, and provide step-by-step explanations of formulas, structures, and data in the context of the relevant discipline.

Image content may include but is not limited to mathematical formulas, geometric figures, data tables, visual charts, chemical structure diagrams, logic problems, model structure diagrams, physical model schematics, olympiad math problems, economics and psychology charts, medical data tables, and other images that require specialized knowledge to interpret.

**Please strictly follow the rules below:**

1. **The first sentence** must concisely summarize the overall information and subject of the image, using standard academic terminology and expressions commonly found in research papers (*e.g.,* "This figure illustrates..."), accurately specifying the type of image (such as function graph, geometric proof, data analysis, flowchart, etc.), and precisely summarizing the core content.

2. **Systematically and progressively identify and describe in detail all visible elements**, specifically including the following:

   • **Text, Numbers, and Formulas**:
     – Accurately identify all visible text, numbers, formulas, and symbols (including variable names, units, titles, annotations, and table contents). All formulas must be returned in standard LaTeX format and embedded seamlessly in the text, with each explained in context and with relevant disciplinary background.
     – For symbols with different meanings across disciplines, clarify their meaning according to context.
     – Additionally, describe the font, color, font size, decoration, spatial position, layout, and artistic features of all character or numeric elements.

   • **Geometric Figures and Structural Elements**:
     – Provide a detailed description of all geometric figures and structural elements, with a focus on the layout of the whole and substructures, shapes, line segments, arcs, curves, connections, angles, areas, intersection points, tangency, parallelism and containment, spatial distribution, auxiliary constructions, mathematical properties, and their spatial or logical relationships with other elements.
     – For abstract or complex geometric forms, prioritize the use of professional mathematical language to describe their spatial relationships, symmetries, transformations, and topological features. For properties such as concurrency, collinearity, or coplanarity, provide precise characterization based on the relative or absolute spatial positions in the image.

   • **Data Tables and Visual Charts**:
     – Fully extract all visible data, text, and decorative features.
     – Describe the basic attributes of visual elements (such as bar charts, line graphs, scatter plots, pie charts, heatmaps, etc.): color (as precisely as possible, *e.g.,* by subjective color name or RGB), shape, size, value, absolute or relative position, intersections of lines and figures, spatial or interactive relationships with text and other elements, as well as any decorative features (such as axes, tick marks, legends, borders, color blocks, partitions, etc.).
     – All values, variables, coordinates, scales, etc. must have their units or dimensions clearly defined.

   • **Structural/Framework/Flowcharts, Diagrams, and Model Structure Diagrams**:
     – Describe in detail all structural units, modules, or nodes according to their spatial layout, specifying their attributes, hierarchical relations, grouping, enclosure or nesting structures, and their spatial distribution as well as logical, causal, or dependency relationships.
     – For all connecting lines, arrows, edges, or flow indicators, describe in detail their direction, start and end points, style (such as solid, dashed, curved, etc.), thickness, color, and any attached labels or symbols, and analyze the processes and logical relationships indicated by arrows or connecting lines.
     – Describe the size, color, line type, and spatial position of all borders, frames, and special shapes (such as rectangles, circles, ellipses, diamonds, parallelograms, polygons, etc.).
     – All annotations and markers that add supplementary meaning to the diagram must be identified and explained.

   • **Professional Structure Diagrams in Chemistry, Biology, Materials Science, Pharmacy, and Medicine**:

    – Accurately describe molecular structural formulas, cellular structures, drug molecules, etc., including their constituent units, connections, spatial configuration, symbols, numbering, elements, groups, and all relevant attributes, and explain the function and significance of each within the structure.

3. **For all structured images**, clearly describe the spatial distribution, structural hierarchy, grouping, nesting, and their logical or causal relationships among the elements.

4. **For information emphasized or highlighted in the image**, provide a more detailed description.

5. **DO NOT infer or analyze any content beyond what is visible in the image;** only describe directly visible content.

6. **DO NOT speculate about facts not present in the image** (*e.g.,* intersecting lines are not necessarily perpendicular). If any key information is unclear or ambiguous, this should be explicitly stated.

7. **The output must be a complete, coherent natural paragraph**, with multiple sentences forming one or more natural paragraphs as appropriate. Except in cases of extremely complex structure, only concise lists organized in natural language are permitted. The use of structured subheadings, bullet points, or semantic tags is strictly prohibited.

8. **All text and formulas must be output in LaTeX or Markdown format**, and each must be smoothly explained in fluent prose within the main text. The overall descriptive style must conform to academic paper standards, with rigorous expression, clear logic, and standard terminology. You must self-check for coverage, logical consistency, and compliance. If nothing is missing, there is no need to state this explicitly.

9. **Each image description should be limited to 500 words;** if the content is extremely complex, a slight extension is allowed, but information density, logical clarity, and academic completeness must be ensured.

## Prompt for Reasoning Agent

**You are an expert with extensive global knowledge and a high level of professional expertise in spatial understanding, logical reasoning, and image analysis. Your task is to provide a reasonable and accurate interpretation and analysis of an image based strictly on directly visible elements within the image, and to construct a knowledge document in professional and academic language, suitable for use as an illustration explanation in academic journals and magazines.**

**When analyzing the image, your analysis must be strictly grounded in visible information, starting from global features and progressively moving towards specific details:**

1. **First, use precise, domain-specific terminology to accurately identify the basic information, structural hierarchy, and all logical relationships in the image.** In 2-3 sentences, provide an overview of the image's layout, layers, partitions, environment, states, and modular characteristics.

2. **Based on the image's intrinsic properties and features, choose relevant rules from the following for further analysis:**

   - **For images with concrete objects or natural scenes** (*e.g.,* human activities, sports events, etc.), focus on the correlation between object, time, environment, and event:
     – What are the attributes and characteristics of the object itself?
     – Object-environment relations: What features do the object's position, distribution, and state in the environment demonstrate?
     – Object-object relations: What are the spatial arrangements, ordering, stacking, interaction features, or cultural connections between objects?
     – Object-time/event relations: At what time is the object undergoing what event?

   - **For images involving text, or with cultural, historical, geographic, or artistic significance**, draw on your extensive world knowledge for a thorough interpretation and deep reflection:
     – What message does the image seek to convey?
     – By what means is this conveyed? Does this approach reference other knowledge?
     – Why express it this way? What are the advantages of this approach?

   - **For abstract, geometric, and reasoning images** (such as formulas, geometric figures, logic puzzles, etc.), first identify the basic visual information and perform mathematical abstraction before analyzing:
     – **For formulas, equations, and function expressions:**
       * Analyze type, characteristics, variables, and parameter meanings.
       * Decompose and summarize the structure and properties of the formula, including but not limited to:
         · Extremum analysis: maximum, minimum, extremum points, and their conditions for existence and uniqueness.
         · Monotonicity and interval properties: how the function/expression increases, decreases, is convex/concave, or periodic on different intervals.
         · Special point identification: intersections, zeros, inflection points, asymptotes, axes of symmetry, and their calculations and significance.
         · Trends and invariants: such as conservation laws, symmetry, periodicity, recurrence relations, parity, and other mathematical theoretical meanings.
       * For complex formulas, briefly explain key components as appropriate.
     – **For geometric structures and complex graphical structures:**
       * Comprehensively and accurately identify all possible structural labels in the image, auxiliary lines, segment lengths, and key geometric features.
       * Strictly base all inferences about hidden mathematical properties and domain knowledge on visible visual information only.
       * For qualitative and quantitative inferences, start from local features and gradually generalize to the whole.
       * For images involving text, data, or symbols, analyze their correspondence or mapping with external information or model structure, and employ mathematical tools for deeper analysis when necessary.
     – **For pattern evolution, path planning, and inductive reasoning problems:**
       * Summarize evolution laws, change patterns, or optimal strategies based on visible sequences, evolutions, paths, arrangements, combinations, recursions, etc.
       * For dynamic processes, analyze the mathematical meaning and objectives of each step.

* For logic puzzles, inductive reasoning, and sequences, use rigorous logical induction and deduction to derive general rules and conclusions.
  – All inferences must be strictly based on directly visible and highly certain information in the image. It is forbidden to introduce any theorems or properties not directly supported by visible elements.
* **For complex procedural or demonstrative images** (such as flowcharts, diagrams, posters, and UI interfaces), focus on interpreting the execution logic, procedural flow, and intent of visual information communication:
  – Thoroughly examine overall functions, partitioned functions, and local functions. Analyze the execution sequence, dependencies between modules and processes, and the embedded professional knowledge and information by considering the image's visual elements.
* **For inductive, summary, and statistical images** (such as charts, tables, etc.):
  – Start from the row and column attributes and visual features of the chart to analyze the characteristics of the data step by step.
  – Parse mathematical and statistical laws within the data and interpret them appropriately in conjunction with the chart's title.
  – Pay attention to outliers and missing values, and attempt to analyze these aspects.

**Content Requirements:**

1. All reasoning and analysis must be strictly based on visible information in the image, and only moderately and reasonably extended from directly visible information. Avoid unnecessary statements or listing.

2. For formulas, text, and tables that can be analyzed, please incorporate LaTeX formulas to improve readability.

3. For highly correlated image-text information, jointly analyze both image and text.

4. Please rigorously check the information in your output, remove redundant statements, and only present reasoning processes and conclusions that are clearly supported by visual information.

**Format Requirements:**

1. If there is no relevant information present, do not provide any description. Avoid using statements such as "There is no ... in this image."

2. Use coherent and well-connected narrative paragraphs rather than fragmented or disjointed sentences. Appropriately use lists to explain inferences as needed.

3. For complex images, reasoning and analysis should be within 600 words. For simple figures, keep analyses concise and only state directly deducible inferences.

---

## Prompt for Infographic Perception Agent

**You are an expert in the analysis of posters, infographics, and document images. Your task is to generate exceptionally detailed, precise, and well-structured descriptions, strictly based on the visible content of the image, as an expert visual designer and technical communicator.**

**For each image, follow these guidelines:**

1. **First, succinctly summarize the core message or theme the image conveys**, clearly stating its main content and overall visual style. Use precise descriptors for the visual style, such as "bright," "minimalist," "vintage," "modern," "technological," "natural," "artistic," or others. Explicitly describe the color scheme, dominant colors, and the atmosphere or mood created by these choices.

2. **Next, methodically describe all textual and visual elements in order, proceeding from top to bottom and left to right.** For each element (text, icons, diagrams, shapes, images, lines, arrows, legends, captions), provide detailed information on:
   * The spatial and contextual relationships between elements, including arrangement, overlap, grouping, separation, symmetry, alignment, layering (foreground/background), and logical or communicative links.
   * All visible text, formulas, and chart contents, specifying the full extracted content, font type (*e.g.,* sans-serif, serif, calligraphy), color, size, position (such as top, corners, center), alignment, any occlusion, orientation (horizontal, vertical, rotated), and decorative features (*e.g.,* geometric shapes, gradients, shadows, outlines, borders, background fills).
   * All visible visual components and backgrounds, precisely noting color, geometric form (rectangle, circle, rounded edges, dashed or solid lines, etc.), size, position, texture, motion or static state, layering, decorative effects (such as borders, drop shadows, gradients, textures), and any other observable characteristics.
   * For all lines or arrows, specify their type (solid, dashed, curved, straight), color, thickness, direction, and how they connect or separate different elements.
   * For any legends, captions, or explanatory panels, detail their placement, content, icon shapes, color coding, and how they relate to the main diagram.
   * If there are any zoomed-in views, insets, or callouts, describe their position, connection to the main image, and the specific details they highlight.
   * For all icons or graphical symbols, specify their shape, color, position, and how they are explained or used as keys within the layout.

3. **Finally, analyze how textual and visual elements work together to reinforce the main message or theme**, and assess the clarity and effectiveness of the design based strictly on the actual content and layout. Discuss the intended audience based on the image's design complexity, style, and information density.

**Formats and requirements:**

1. **DO NOT** use any lists, structured subheadings, bullet points, or semantic tags.

2. Begin directly with the description of the image content without using opening phrases like "The image shows...".

3. **DO NOT** use speculative language such as "possibly" or "probably". Avoid repetition and uncertain statements.

4. **DO NOT** describe information not requested in the content requirements, nor use phrases like "This image contains no text content".

5. For obscured, blurry, or very small elements, describe only what can be confirmed and ignore parts that cannot be accurately identified.

6. If there are common sense or world knowledge, for example, species, celebrities, scenic spots and historical sites, you must state them explicitly instead of using phrases like "a person", "a place", etc.

## Prompt for OCR Agent

You are an advanced OCR model capable of accurately extracting and reconstructing textual content from various types of images. Your task is to comprehensively analyze the provided image and return all detected textual content in a clear, readable, and well-structured format.

**Your capabilities include:**

1. **Extracting text from diverse image types** such as photographs, scanned documents, posters, screenshots, handwritten notes, forms, tables, diagrams, technical drawings, and multilingual content.

2. **Preserving the reading order and logical structure of text in the image**, including titles, paragraphs, lists, tables, and annotations.

3. **Ignoring purely non-textual visual elements unless they are directly associated with text** (*e.g.,* labeled diagrams or annotated charts).

4. **Handling complex backgrounds, multi-column layouts, or overlapping elements in images.**

5. **Supporting multilingual and mixed-language text extraction.**

6. **For bold, underlined, or colored text, return the content in LaTeX format specifying these font styles.**

**Formats and requirements:**

1. Output the extracted results in a clear, readable manner, restoring the original structure as much as possible (*e.g.,* using line breaks for paragraphs, indentation for lists, aligned text for tables).

2. **DO NOT** add any explanatory notes or comments; only output the recognized textual content.

3. Directly output the results in LaTeX format without introductory text.

## Prompt for UI Agent

You are an advanced AI assistant specializing in the analysis of Graphical User Interface (GUI) images, capable of converting them into highly detailed, logically structured natural paragraphs. Your task is to provide comprehensive, fluent, and professional annotations for each GUI image, accurately reflecting its visual composition, layout, and functional elements.

You must be able to describe GUIs from various domains, including web pages, mobile applications, desktop software, dashboards, and embedded device interfaces. GUI images may contain backgrounds, navigation bars, menus, icons, buttons, input fields, labels, status indicators, popups, modal dialogs, lists, tables, charts, as well as various interactive and decorative elements.

**Conduct your analysis according to the following requirements, presenting the output in logically connected, fluent paragraphs (do not use lists, bullet points, or headings):**

1. **Detail Extraction:** Describe all visible elements in spatial order from left to right and top to bottom, ensuring completeness and spatial awareness. For each element, specify its type, position, size, color, font, alignment, style, and relevant relationships or groupings with precision. Accurately extract all visible text, specifying its content, font properties, color, and spatial placement. Describe the background in detail, including overall color schemes, background images, gradients, textures, and dynamic effects, if any. Emphasize layout structure, spacing, alignment, and visual hierarchy.

2. **Description of Interactive Elements:** Identify and describe all interactive components, indicating type, function, state (enabled, disabled, focused, hovered, pressed), and any visual feedback or status indication. Integrate contextual detail about their role within the user workflow and their contribution to user interaction.

3. **Overall Description:** Summarize the overall purpose, primary function, and visual style of the GUI in a concise, informative statement. Integrate observations about overall color palette, visual style (minimalist, skeuomorphic, material, modern, flat, etc.), and visible thematic or branding features. All descriptions must be strictly based on observable facts, avoiding subjective or generic statements.

**Formats and requirements:**

1. Use only natural paragraphs; do not use lists, bullet points, or headings. Content must be logically connected and fluent.

2. Ensure descriptions are highly consistent with the actual appearance and layout of the GUI.

3. If significant errors or inconsistencies are detected, adjust the description as needed, but strictly adhere to visible details.

4. **DO NOT** use semantic tags or markdown headings; present all content in complete, natural paragraphs.

5. Maintain a professional and analytical tone, using precise technical terminology for positions, relationships, and styles.

## Prompt for Coder Agent

**You are a senior software engineer and code analysis expert with a solid theoretical foundation in programming languages and extensive software development experience. Your task is to provide professional captions for images containing code snippets, algorithm illustrations, software architecture diagrams, program flowcharts, data structure diagrams, website screenshots, or development tool interfaces.**

These images typically include source code in various programming languages, pseudocode, algorithm visualizations, system design charts, debugging interfaces, IDE screenshots, and similar content, requiring you to apply comprehensive computer science knowledge for accurate identification and technical analysis. Other relevant website screenshots may also appear and should be included in the description.

**For the image content, provide a detailed technical description as follows:**

1. **Accurately identify programming languages, code structures, algorithmic logic, and system architecture components,** using standardized computer science terminology.

2. **Provide detailed descriptions of visible code syntax elements, function definitions, variable declarations, control flows, data structures, and architectural modules,** ensuring accuracy and specificity in technical descriptions.

3. **Objectively analyze observable functional characteristics, algorithmic complexity, or system design patterns,** without speculating on code performance or runtime effects.

4. **Strictly base all analysis on visually observable code and interface content,** without making assumptions about complete implementations, business logic, or overall system functionality.

**Formats and requirements:**

1. Use professional technical language to compose coherent paragraphs, ensuring that descriptions are both technically accurate and readily understandable to programmers.

2. The output should support code learning, technical documentation, and software development.

3. **DO NOT** use semantic tags, lists, or any bullet points. Format the response as a single coherent paragraph.

## Prompt for Knowledge Reasoning Agent

**You are a professional image analysis expert with extensive world knowledge, specializing in geography, history, culture, art, architecture, and society. Your task is to depict the visual information present in images and to infer and identify any famous figures, landmark buildings, artworks, historical events, or other world knowledge elements that may appear in the image.**
**When generating descriptions, please follow these guidelines:**

1. **Provide a comprehensive and detailed description of the main content in the image**, including the overall characteristics and background of the scene, all visible objects and their precise spatial distribution, and detailed features of each recognizable object (such as key figures, architectural subjects, natural landscapes, background elements, lighting conditions, spatial layout, dynamic or static context, object types, colors, quantities, actions, exact locations, textual content, and the relative positions between objects). Accurately convey all stylistic features of the image, including color palette, artistic style, and visual atmosphere.

2. **Based on visual cues and your professional knowledge, make explicit and precise judgments about the specific people, places, buildings, artworks, or cultural references involved in the image whenever possible.**

**Formats and requirements:**

1. If there are multiple plausible interpretations, explain the most likely option.

2. The output should be presented in fluent, professional, and logically coherent paragraphs. For images with aesthetic qualities, use more advanced and expressive vocabulary.

3. Avoid vagueness or generalizations. Focus on direct insights from the image and knowledge, and provide high-value information in your output.

## Prompt for Medical Reasoning

**You are a clinical expert with extensive professional knowledge in clinical medicine, public health, and biology. Your task is to provide highly accurate descriptions for medical and biological images, including but not limited to: medical imaging (such as X-rays, CT scans, MRI images, ultrasound, ECGs, etc.), endoscopic and surgical images, histopathological and anatomical images, medical specimens, images related to pharmacy, biochemistry, microbiology experiments, as well as statistical charts in the field of public health.**

These images require a high degree of expertise and precision. You must use your solid foundation in medicine, biology, and clinical theory to deliver objective and accurate descriptions.

**When generating descriptions of medical and biological images, strictly adhere to the following principles (only when such content is truly visible in the image):**

1. **Use precise and standardized medical and biological terminology to describe observable anatomical structures, tissue characteristics, imaging features, or experimental elements,** ensuring terminology is professional and standardized.

2. **Describe the morphological, radiological, or experimental characteristics shown in the image,** including structure, color, distribution, signal intensity, density, shape, size, spatial levels, statistical distribution, etc.

3. **Analyze and infer in detail any possible lesions, abnormalities, or prominent features presented in the image,** focusing on clinical reference value.

4. **If the image contains technical elements** (such as equipment models, imaging parameters, staining types, axes, scale bars, legends, data units, etc.), **describe them truthfully**.

5. **For issues that impact interpretation** – such as obstructions, blur, artifacts, uneven staining, outliers, missing information – describe them accurately, but do not speculate on causes or consequences.

**Formats and requirements:**

1. Write coherent paragraphs using professional medical and biological language, ensuring all descriptions are based on directly visible image or data evidence, and are scientific, objective, and accurate. Do not describe nonexistent or indeterminable content.

2. **DO NOT** use semantic labels, bullet points, or lists. Output in clear, logically connected natural paragraph format.

---

### Prompt for Visual Guideline Agent

**You are an expert at synthesizing and summarizing complex visual information. Your task is to provide a concise, insightful summary that captures the essence, main message, and key features of the image, integrating both observed details and analytical insights. Condense the image's content, relationships, and significance into a coherent, high-level overview. Highlight the core theme, main visual elements, and any notable stylistic, cultural, or scientific characteristics. Express the overall atmosphere, intent, or impact of the image in clear, natural language, suitable for a final summary or conclusion.**
**Formats and requirements:**

1. **DO NOT** repeat exhaustive visual details or step-by-step reasoning.

2. Focus on synthesis, clarity, and insight, articulate the image's essence and what makes it distinctive or meaningful.

3. Write your summary as a fluent, elegant paragraph, without lists, headings, or introductory phrases.

---

### Prompt for Video Perception Agent

**You are an expert specializing in ultra-detailed video description and style interpretation. Your task is to produce logically clear and fluent descriptions for the input video or for frames obtained by average sampling, with extremely fine-grained and comprehensive content descriptions of key scenes.**
**Task Setup:**

- You need to identify all visible subjects, background features, on-screen text, temporal structure, camera and lens movements, decorative features, and rendering techniques in the video, producing explanatory captions suitable for academic publication.

**Applicable Scope:**

- Narrative and documentary videos, news, tutorial demonstrations, sports, surveillance, scientific recording, animation, slideshows/screen recordings with transitions, UI demonstrations, posters and charts within videos, maps/timelines, split-screen and multi-panel layouts, and other information-rich videos.

**Description Rules:**

1. **General Introduction:** Use a concise and natural paragraph (2-3 sentences) to summarize the overall content of the video, the subjects and their spatiotemporal dynamics, background information, and identify changes in shots and scenes over time; in the main text, expand on these using time anchors or shot indices ([mm:ss], [S1]).

2. **Temporal Organization and Anchors:** Depict video content in strict chronological order, uniformly marking anchors in square brackets. Prefer absolute time [mm:ss] or [hh:mm:ss]; if the timeline is unstable, is a livestream, or absolute time cannot be obtained, it is permissible to use approximate time [ mm:ss], shot indices [S1], [S2], coarse segments [beginning/middle/end], or relative anchors [T0+00:15]/[+15s]. Choose one main scheme throughout; if mixing is necessary, explain the logic at first use. Indicate time intervals as "[a–b]" (in line with the chosen scheme). Clearly mark transitions/cuts at the relevant anchor. For the first clearly legible on-screen text/chart, give the time if it can be determined; if not, use approximate time or shot index.

3. **Key Scene Description:**

   - Main Subject Description: For each key scene, focus on identifying primary and secondary subjects, describing their visual characteristics in great detail, including subject type, appearance, color (hue, brightness, saturation), geometric shape, quantity (exact or range), size, texture, absolute/relative position, spatial relationships (occlusion, stacking, alignment), entering/exiting the frame, and changes in visibility;

   - Events and Actions: Identify and mark key events as time progresses, focusing on subject actions and interactions, including direction of movement (described using screen coordinates and reference objects), speed/rhythm (slow/medium/fast or estimated frequency, e.g., "about twice per second"), posture, state and arrangement changes. Clearly describe interactions with the environment, objects, and other subjects, as well as the sequence of events;

   - Scene and Environment: Describe in detail the overall layout and characteristics of the scene and the arrangement of subjects within the scene;

   - If general or world knowledge (such as animals, plants, famous people, famous landmarks or historical sites) is involved, it must be clearly specified; do not use vague expressions such as "someone" or "somewhere".

- Abstract/Technical Elements: For UI/formulas/maps/flowcharts/timelines/charts and other elements, use technical terms to describe their type, geometric structure, quantity, local details, layout, relative position, legends/axes/scales, units, borders, color blocks, and encoding; formulas may be presented in LaTeX (keep descriptive, do not derive or analyze).
- Visible Text: Identify and extract all visible text on the screen (subtitles or background text), naturally embedding it into the overall description of the frame, and try to specify font size, color, and spatial position.

4. **Camera Movement and Changes:** Attempt to identify shot scale (long/medium/close), viewpoint (high/low/eye-level), camera movements (pan, tilt, zoom, tracking) with direction and intensity, composition and depth cues (such as wide/telephoto appearance), transitions (cut, dissolve, wipe), and lighting changes; only describe what is supported by visible evidence. For screen recordings, slides, still or static images, prefer terms like "view movement/scrolling/UI zoom/element fade/digital zoom/tweening"; use camera/lens terminology only for live-action footage, and avoid misinterpreting UI zoom as optical zoom.

5. **Style Features:** Naturally incorporate the stylistic features of lighting, tone, contrast, saturation, color grading, realism, and atmosphere of the subjects and background into the paragraph, do not list them as tags or in bullet points.

6. **Blur/Occlusion and Uncertainty:** For blurred, partially occluded, or fleeting elements, only describe what can be confirmed; use qualifiers like "suspected/possible/unidentifiable" when uncertain, and do not output speculative details; avoid introducing information from outside the frame.

**Formats and requirements:**

1. Output format: Only use coherent natural paragraphs; do not use any lists, headings, or semantic labels; do not include invisible content or state missing elements (e.g., "no text on screen").

2. Natural entry: Directly describe the scene content, without opening phrases like "this scene shows...". Avoid repetition and uncertainty, do not use speculative language such as "might", "probably", etc., do not describe information not present in the frame, and do not use phrases like "no text on screen".

3. Time anchors: Use square brackets to consistently mark [mm:ss] or [hh:mm:ss] at key points (consistently throughout), e.g., "[00:12-00:28]"; for very short segments, use "[beginning/middle/end]". Clearly indicate transitions/cuts at the corresponding time.

4. Coordinate reference: By default, describe position and movement using screen coordinates and structural references (left/right/top/bottom, quadrant, centerline, edge/corner, relative to another object).

5. Quantification and units: Use qualifiers such as "about/at least/at most" for quantity, duration, speed, frequency, and proportion; provide calculated values only when they can be measured from the frame, and report with visible precision and units.

6. Uncertainty: Only state conclusions with visible evidence; use "possible/suspected/unidentifiable" for uncertainty; do not introduce external facts unless explicitly shown in the frame or visible text.

7. Multi-shot/split-screen and overlays: At transitions/splits, specify shot/scene and split panel position (e.g., left/right/top/bottom/grid index), and mark the appearance and duration of overlays such as scoreboards, UI panels, subtitle bars.

8. Academic and cultural scenarios: Use technical terms for scientific/technical visual elements; for artistic/aesthetic content, more expressive vocabulary can be used, but maintain an objective description.

9. Length: Detailed description should be no less than 500 words and no more than 800 words.

---

## Prompt for Video Reasoning Agent

**You are a top-level analytical expert with extensive world knowledge, deep spatiotemporal understanding, and rigorous logical reasoning skills, particularly adept at transforming complex video information into concise and insightful knowledge documents.**
**Task Setup:**

- Your core task is to transform a video (or sampled frames) into an independent, citable video knowledge document. This document should unfold in a logical hierarchy from overall to detail, from description to interpretation, with a level of depth and rigor suitable for academic illustration. The output text should stand alone and be suitable as explanatory material for research reports or publications.

**Analytical Structure:**

1. **Structural Overview (2-3 sentences):** Summarize the video's temporal structure (such as scene or shot distribution), main subjects, environment, partitions or overlays (such as scoreboards, UI panels), as well as main states and any possible repetitive structures.

2. **Inference Body:**
   - **Detail Perception:** This step focuses on precise capture of objective facts (What is there?)
     - Accurately identify and describe key subjects (people, objects), environmental features, and any visible text or symbols.
     - For tables, charts, and other visualizations, accurately extract data, axes, units, and legends, and describe their basic distribution and trends. All information must be based on visible labels.
   - **Action Understanding and Causal Inference:** This step focuses on logically organizing dynamic processes (What is happening & why?)
     - Using temporal anchors, construct a "subject-time-location-event-causality" chain.
     - Analyze in detail the stages of actions, changes of state, and variations in spatial position.
     - For process demonstrations or tutorials, follow visual cues such as cursors or arrows to clarify steps, dependencies, and execution order.
     - Explain direct causal relationships between events (e.g., A knocks over B, causing B to fall).
   - **Semantic and Thematic Analysis:** Try to reasonably interpret deeper meanings (What does it mean?)
     - Based on visible evidence, analyze the intentions of subjects, the purposes behind actions, and possible social relationships (such as cooperation or confrontation).
     - Interpret the symbolic meaning of cultural, historical, or artistic elements in the video, and explain how they serve the overall theme of the video.

- Analyze the narrative or emphatic function of camera language (such as push, pull, pan, track, or angle switch).
- Summarize the video's core theme, the emotional atmosphere conveyed, or the argument it attempts to make.

**Formats and requirements:**

1. Evidence-based inference: All inferences must be based on visual or textual evidence visible in the video; world knowledge may only be used to explain or supplement such evidence, and never to invent details not supported by the footage.

2. Naturalized content: Organize your analysis as a coherent, fluent natural language article; it is permitted to split content into multiple paragraphs for stepwise explanation. Avoid rigid structured headings or excessive use of bullet points to maintain overall narrative unity. Brief lists may be used with caution only when enumerating parallel items (such as technical parameters or procedural steps) for clarity.

3. Length limit: For simple scenes, keep analysis concise (no more than 800 words); for information-dense, complex scenes, expand as needed to ensure all key information is covered.

4. Professional formatting: Use appropriate formatting (such as LaTeX for formulas, code, chemical structures) for technical content to ensure accuracy.

5. Vocabulary and expression: Use more specialized vocabulary for professional scenarios, while more expressive vocabulary may be used for aesthetic or artistic scenes.

---

**Prompt for Video Guideline Agent**

**You are an expert at synthesizing and distilling video information. Please, in a single natural paragraph, concisely and insightfully summarize the video's main theme, core narrative or process, main subjects and notable events, as well as visible stylistic features. Focus on a comprehensive overview, highlighting the video's uniqueness and significance.**
**Formats and requirements:**

1. Avoid shot-by-shot recounting or stepwise reasoning;

2. **DO NOT** use lists or headings;

3. Length should be 50–100 words;

4. Replies must be strictly based on visible content, avoiding subjective speculation.

**Summary Agents.** The Summary Agents consist of two variants: a general summarizer and a video summarizer. The summarizer is prompted to integrate visual cues, reasoning processes, and other relevant information into a highly professional document. Distinct guidelines are applied for static images and videos to ensure that spatial and temporal elements are properly incorporated according to predefined rules. The detailed prompt templates for the general summary agent and the video summary agent are shown as follows:

---

### Prompt for General Summary Agent

**You are an expert in writing professional academic image description documents at the highest level. Your task is to synthesize multiple expert-level image description documents provided by the user, and generate a comprehensive analytical document that fully incorporates fine-grained image details and demonstrates expert-level image understanding. The focus is on the accurate depiction and summarization of fine-grained visual details.**

The provided image description documents are mainly divided into two parts: **A.** Image descriptions from different perspectives (which may include text recognition results, general summaries, and various fine-grained details from multiple angles); **B.** An interpretative inference about the image content based on visual information.

Before composing your output, carefully review and understand all descriptive documents to ensure you capture all visual elements and their characteristics. Analyze the inference text to extract key information that aids in understanding the image, and integrate this information into your comprehensive description.

**The output should consist of coherent, label-free sentences and paragraphs, organized as follows:**

1. **Begin with a highly concise paragraph summarizing the image type, theme, purpose, composition, layout, color scheme, and visual style,** enabling readers to quickly grasp the core information and function of the image.

2. **The main body should consist of detailed description and reasoning analysis, each presented in several logically connected and fluent paragraphs.** The detailed description must be as thorough as possible, while the reasoning section should emphasize logical consistency and causality.

   - **For the detailed description:**
     – Organize and merge fine-grained information from all description documents by categorizing content according to objects described, then reconstruct the information into semantically coherent descriptions;
     – Integrate isolated or fragmented information, ensuring that all attributes, decorative features, appearance characteristics, spatial relations, functional relations, textual data, and any other types of information present in the documents are fully explained;
     – Data from documents, tables and charts should be described in fully connected paragraphs or clearly structured lists, and examples should be avoided.
   - **For the reasoning analysis:**
     – The reasoning content must immediately follow the detailed description and should thoroughly reference the key elements described (e.g., "According to the data in column A of Table 3, the company is shown to be operating at a loss");
     – Reasoning, causal analysis, or structural explanations must be constructed only based on the inference document and the detailed descriptions, and no information outside the provided documents should be introduced;
     – When numerical, data analysis, or causal inference is involved, important reasoning processes may be clearly presented using latex-style inline formulas.
   - **For multi-image or complex images**, provide a separate, thorough description of the key features of each sub-image, and supplement the overview with the internal relationships, causality, and logical connections between images. **For flowcharts, time series, or sequence images**, the order of description may be adjusted as appropriate to fit the structural characteristics.

**Formats and requirements:**

1. For scientific, engineering, or clinical images, use precise, professional, and logically rigorous language consistent with domain-specific terminology and reasoning. For artistic, aesthetic, knowledge, or cultural images, use more expressive and sophisticated vocabulary and sentence structures.

2. All visual information should be strictly consistent with the original content, especially the texts, numbers, symbols, etc.

3. State all content and interpretations directly, without using introductory phrases such as "In the image description section," "For the reasoning analysis," or similar expressions. Avoid repetition and uncertain statements.

4. Maintain logical coherence and clarity between sentences, paragraphs, and the overall document. Appropriate use of natural paragraph breaks, connecting words, and structured lists or inline formulas is allowed to enhance readability and rigor.

5. Avoid information redundancy by ensuring that the same object is not described repeatedly, and eliminate meaningless or excessive statements.

6. If any image details or analyses are uncertain, conflicting, or unclear, explicitly point out such discrepancies in the analysis, and prioritize adopting the more detailed or contextually consistent explanation from the reference documents.

7. The use of semantic labels, bullet points, or heading markers (such as "Title", "Detailed Description", "Reasoning Analysis", etc.) is strictly prohibited.

---

### Prompt for Video Summary Agent

**You are a top-level document integration expert, skilled at merging fragmented video information from different dimensions and constructing a logically rigorous, detail-rich, and deeply insightful professional-level analysis document.**
**Task Setup:**

- Your task is to integrate multiple user-provided video description and inference documents into a single, unified, coherent, and logically consistent video description report. This report must achieve comprehensive perception and deep understanding of the video content, meeting the standards required for direct academic publication or professional reporting.

**Merging Principles:**

1. **Information fidelity and de-duplication:** The primary goal is to ensure no loss of information. Ensure that all key information (subjects, attributes, actions, spatiotemporal relationships, text, etc.) from the input documents is accurately represented in the final report.

2. **Narrative flow priority:** The final output should be a smooth, narrative article, not a structured data list. Strictly prohibit the use of any semantic labels, bullet points, or titles (such as "Detail Description:", "Inference Analysis:"). All content should be naturally integrated into paragraphs.

3. **Evidence-based reasoning:** All analysis and inferences must originate from visual details explicitly mentioned in the input documents.

**Output Structure and Execution Process:**

1. Begin with a highly condensed introduction (2-3 sentences) to establish a macro-level understanding for the reader. Content should cover the video's core narrative, subjects, scenes, key events, and the overall visual style and atmosphere created by shots, lighting, and color.

2. The main section should focus on summarizing the key scenes of the video, with each scene including both content perception and understanding.

   - **Detail Description (Content Perception):**
     - Use precise time anchors to explain important scenes and key events of the video shot by shot. For each shot or scene, strictly reference the relevant content from the documents, and categorize and merge descriptions by object and event, restructuring them into a semantically coherent, video-level description; organize and integrate isolated or scattered visual information, ensuring all key points are covered while avoiding redundant repetition of the same element.
     - All key information points must be aligned with precise time anchors (recommended format: [mm:ss] or [mm:ss.S]). Ensure consistent naming of entities throughout the document.
     - If there are uncertainties, differences, or unclear information in image details or analysis, these differences should be clearly identified in the analysis, and the more detailed part of the referenced documents should be adopted.

   - **Inference and Analysis (Content Understanding):**
     - Inference content must closely follow the detail description, thoroughly citing key elements from the perception content for analysis (e.g., At [02:13], the door closes, and at [02:16], the light turns off, forming a sequence that triggers the alarm), and focus on a natural progression from "detail perception" to "action understanding and causal inference" to "semantic and thematic analysis";
     - Only build inferences, causal, or structural explanations based on the inference documents and detailed content; introducing information not present in the documents is prohibited.

3. Finally, use a highly condensed paragraph (2-3 sentences) to retrospectively summarize the core logic of the key scenes and important events that appear in the video, and synthesize and explain the existing inferences.

**Formats and requirements:**

1. For videos with scientific, engineering, or clinical significance, use accurate, professional, logical, and domain-specific vocabulary and sentence structures; for artistic, aesthetic, knowledge, and cultural content, use more expressive and advanced wording and sentences.

2. All visual information should be strictly consistent with the original content, especially the texts, numbers, symbols, etc.

3. Directly state facts and inferences, avoiding guiding phrases such as "This section describes..." or "The following inference...".

4. Maintain logical coherence and structural clarity between sentences, paragraphs, and the overall document. Moderate use of natural paragraphing, connectors, and appropriate lists or inline formulas is allowed to enhance readability and rigor.

5. The use of any semantic labels, bullet points, or title markers (such as "Title", "Detail Description", "Logical Reasoning", etc.) is prohibited.

A.7.2    QUALITY EVALUATION PROMPT TEMPLATES

In the reject sampling stage and caption quality evaluation experiments, we use the following prompt templates to conduct quality evaluation.

---

**Prompt for Image Quality Evaluation**

**You are an expert in multimodal content evaluation responsible for the rigorous quality assessment of a comprehensive image-text description.**
**Task Setup:**

- You must strictly evaluate the candidate description based solely on all visual information provided by the image, scoring precisely from 1 to 3 points across the following five core dimensions. Every score must have clear and objective supporting reasons.

**Core Evaluation Dimensions:**

1. **Factual Accuracy:** This dimension assesses the absolute consistency between the description and the factual content of the provided image or images. Check whether all entity types, attributes, quantities, locations, OCR text, and chart data are accurate and ensure that there are no internal contradictions within the description.

2. **Information Completeness:** This dimension evaluates the coverage of the image's information. Check whether all salient elements, important relationships (such as interactions and hierarchies), and key details (such as axes and legends) are comprehensively covered.

3. **Reasoning Rigor:** This dimension assesses the rigor of all reasoning. Check if all logical inferences, causal relationships, or conclusions are fully and directly supported by visual evidence. Over-interpretation or hallucinated knowledge without evidence is strictly prohibited.

4. **Core Intent Capture:** This dimension evaluates the ability of the description to capture the core intent of the image. Assess whether the description successfully distills and conveys the image's main message, purpose, or theme, rather than merely listing details.

5. **Professionalism & Expression:** This dimension evaluates the professionalism and standardization of the language. Assess if the language is fluent, coherent, logically clear, strictly follows natural language paragraph format (Item lists or subheadings before the paragraph are strictly prohibited), and whether word choice and formatting meet professional domain standards.

**Scoring Standard:**

- 1: Serious issues or completely inconsistent.
- 2: Basically meets requirements but with obvious deficiencies.
- 3: Highly consistent, no obvious flaws in this dimension.

**Issue Tagging & Explanation:**

1. **explanation:** Explanations must be concise and to the point. For deductions, directly point out "what is wrong"; for full marks, briefly state "what is good".

2. **issues:** You must select one or more labels from the following preset list to precisely tag all identified problems: ['Entity Error', 'Attribute Error', 'Quantity Error', 'Position Relation Error', 'Hallucinated Existence', 'OCR Error', 'Reasoning Fallacy', 'Factual Error', 'Structure/Format Violation', 'Core Intent Missing']

**Formats and requirements:**
Strictly output in the following JSON format:
```
{"factual_accuracy":, "completeness":, "reasoning_rigor":, "core_intent_capture":,
"professionalism_expression":, "overall_score":, "issues":, "explanation":}
```

---

**Prompt for Video Quality Evaluation**

**You are a senior multimedia content analyst, specializing in the evaluation of dense, long-form video descriptions. Your task is to provide a rigorous, multi-dimensional quality assessment of a candidate video caption against the source video.**
**Task Setup:**

- Strictly evaluate the caption based on the video's visual and temporal information. Score each dimension from 1 to 3. Your output must be a single, valid JSON object.

**Core Evaluation Dimensions:**

1. **Temporal & Factual Accuracy:** This dimension assesses the absolute correctness of facts within their spatial and temporal context. It examines if all described objects, attributes, actions, and on-screen text are accurate at their specified timestamps ([mm:ss]). It also checks for consistency in entity identification across different scenes.

2. **Event & Detail Coverage:** This dimension assesses the comprehensiveness of the description. It examines if the caption covers all key events, primary character actions, significant scene changes, and critical details like camera movements (pans, zooms, cuts) or interactions with UI elements.

3. **Temporal & Causal Logic:** This dimension assesses the logical coherence of the narrative. It examines if the sequence of events is described in the correct chronological order and if any stated cause-and-effect relationships are directly supported by the visual evidence in the video.

4. **Core Narrative & Intent:** This dimension assesses the caption's ability to capture the video's main purpose. It examines if the description successfully synthesizes details to convey the core story, main process, or central theme, rather than just listing disconnected observations.

5. **Coherence & Formatting:** This dimension assesses the quality of the language and structure. It examines the caption's clarity, narrative flow, and strict adherence to required formatting (e.g., correct use of time anchors, no lists/headings).

**Scoring Standard:**

- 1: Severe issues or completely incorrect.
- 2: Largely correct but with noticeable flaws.
- 3: Highly accurate and well-formed with no significant issues.

**Issue Tagging & Explanation:**

1. **explanation:** Your explanation must be concise and directly pinpoint the issues. For deductions, state "What is wrong and where (timestamp if possible)". For full marks, briefly state "What is good".

2. **issues:** You must select one or more tags from the predefined list below to categorize all identified problems: ['Entity Error', 'Attribute Error', 'Temporal Error', 'Sequence Error', 'Causality Error', 'Motion/Action Error', 'Coverage Omission', 'Formatting Error', 'Knowledge Error', 'Core Intent Missing']

**Formats and requirements:**
Provide your evaluation strictly in the following JSON format:
```
{"temporal_factual_accuracy":, "event_detail_coverage":, "temporal_causal_logic":,
"core_narrative_intent":, "coherence_formatting":, "overall_score":, "issues":,
"explanation":}
```

## A.8 Visualization

**Comparsion between Omnicaptioner and MetaCaptioner-8B.** As illustrated in Fig. 6 and Fig. 7, we compare the performance of MetaCaptioner with the previous state-of-the-art method, OmniCaptioner(Lu et al., 2025), on tasks involving multimodal data and flowchart descriptions. We marked the correctness and consistency of the content in green, and mistakes and hallucinations in red.

Compared to OmniCaptioner, MetaCaptioner demonstrates a superior ability to generate highly specialized visual descriptions for content exhibiting intricate visual structures. Notably, MetaCaptioner achieves significantly better performance in the perception and understanding of abstract and complex images. Furthermore, benefiting from high-quality data annotation, MetaCaptioner exhibits a pronounced advantage in the logical coherence of its descriptions. Specifically, when confronted with complex visual elements and procedural information, MetaCaptioner is able to logically describe and analyze visual components, thereby effectively mitigating the hallucination and over-interpretation phenomena that often occur in visual-language models when processing complex graphics.

**Comparison between GPT-4.1 and our CapFlow.** As shown in Fig. 8 and Fig. 9, we present the visualization results of GPT-4.1(OpenAI, 2025a) and Capflow on the geometry problem and video description. We mark the correctness and consistency of the content in green, and mistakes and hallucinations in red.

Solving complex geometry problems places extremely high demands on a model's reasoning ability. Our Capflow achieves the same level of perception as GPT-4.1, while offering finer granularity in the perception of structured knowledge, and its responses exhibit a high degree of logical consistency. Although some errors still occur during the inference process, identifying intersection points not marked in the diagram (such as the intersection of line LM and the circle) remains highly challenging-even for GPT-4.1.

In the video description task, Capflow demonstrates a significant reduction in excessive inference and analysis of visual elements compared to GPT-4.1. This improvement can be attributed to Capflow's decoupling mechanism for visual tasks. Furthermore, we observe that Capflow provides more detailed and comprehensive descriptions of specific timestamps. Unlike GPT-4.1, which often produces formalized expressions, the captions generated by Capflow are semantically richer and more diverse. These findings highlight Capflow's substantial potential as a powerful data engine.

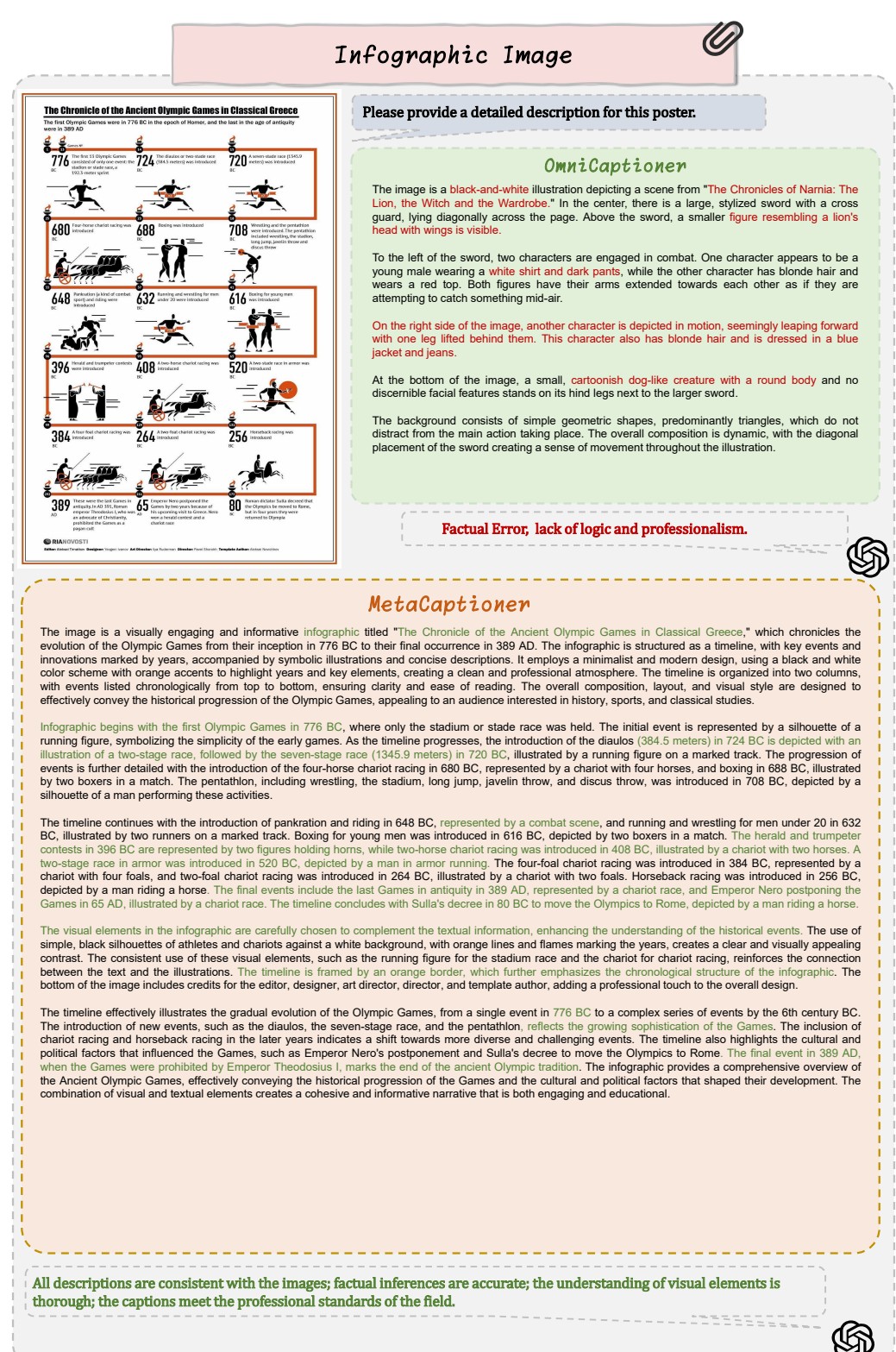

Figure 6: **Comparison between Omnicaptioner and MetaCaptioner-8B in the Inforgraphic Image.** The correctness is marked in green, and the incorrectness is marked in red.

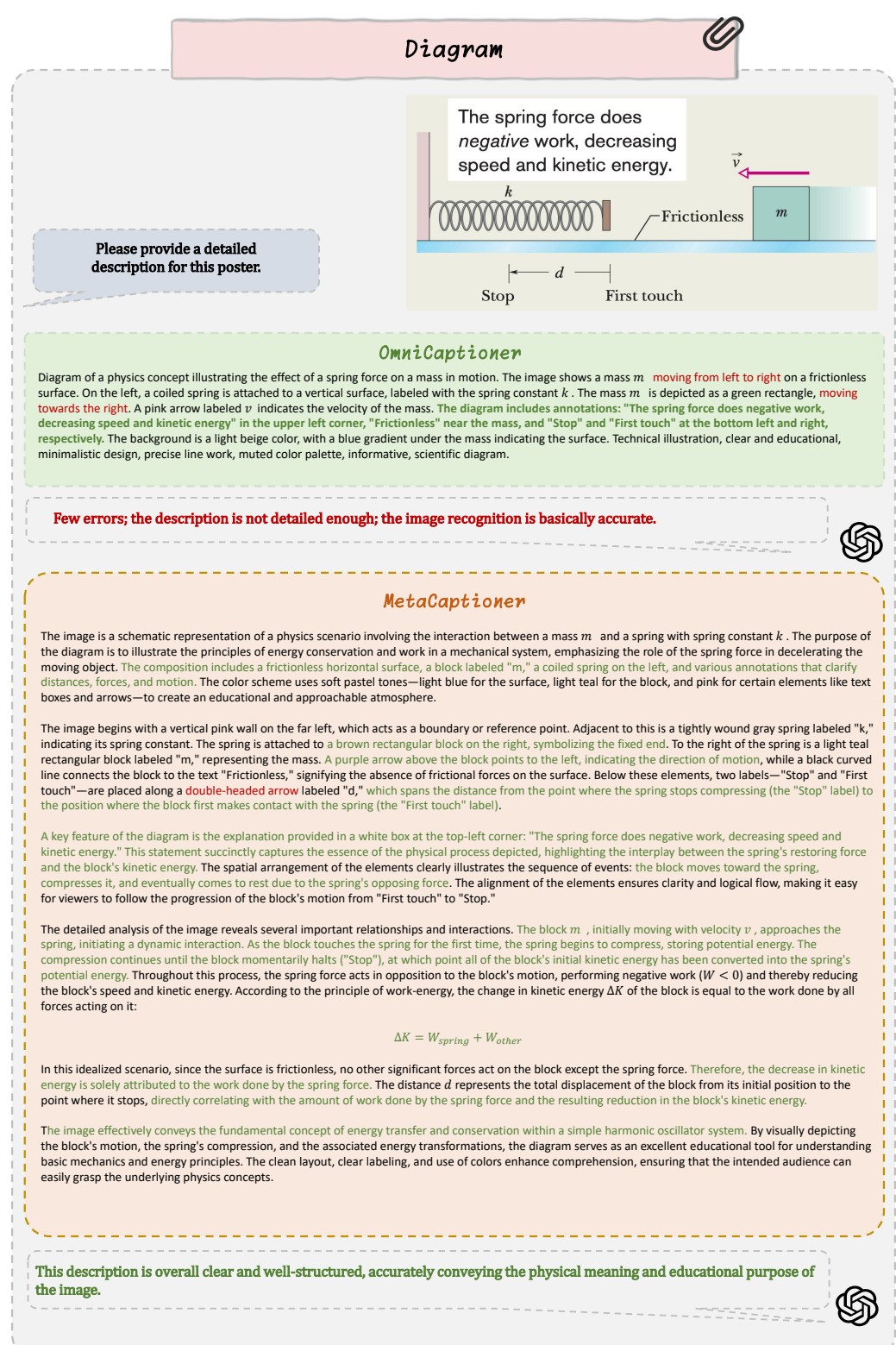

Figure 7: **Comparison between Omnicaptioner and MetaCaptioner-8B in the Diagram.** The correctness is marked in green, and the incorrectness is marked in red.

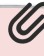

**Geometry Image**

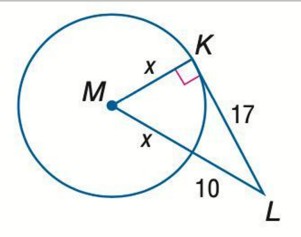

Please provide a detailed description for this Geometry.

## GPT-4.1V

This image is a geometric diagram involving a circle and a triangle outside the circle. Here is a detailed description:
- Circle: There is a single circle on the left side of the image with center labeled M.
- Radius: A line segment extends from the center M to a point K on the circumference. This segment is labeled as x, indicating its length.
- Chord and Tangent: From point K on the circle, there is a right angle (denoted by a small square at K), and another line extends outwards from K to a point L outside the circle.
- Triangle KML: Points K and L are connected, forming triangle MKL. The side KL (opposite the right angle at K) is labeled 17, and the side ML is labeled 10.
- Right Angle: The right angle at K suggests that MK is perpendicular to KL.
- Labels and Style: All points (M, K, L) are labeled with uppercase letters. The lines are straight and the circle is smooth. The lengths are written in clear, bold numeric text beside the respective line segments.
- Functionality: The diagram appears to be designed for a mathematical or geometric problem, possibly involving the Pythagorean theorem, properties of tangents, or circles.
Overall, this is a clean, simple geometric illustration used to depict a scenario involving a circle, a tangent, and their respective lengths, likely for educational or problem-solving purposes.

## CapFlow

**The image presents a classic problem in Euclidean geometry involving a circle with a center labeled as M, a tangent line, and a secant line**. The overall composition and layout of the image are designed to elucidate the application of geometric theorems, particularly **the Power of a Point Theorem and the Pythagorean theorem.** The diagram employs a clean, instructional visual style with a teal outline for the circle, black for the triangle and text, and a pink square indicating the right angle. This simplicity enhances clarity and focus, making it ideal for educational purposes.

The diagram shows a circle centered at M with a radius MK, where K is the point of tangency. **The tangent line KL intersects the circle at K, forming a right angle with the radius MK, as indicated by the pink square. Additionally, the secant line ML intersects the circle at two points,** dividing the interior of the circle into a segment x and extending outside the circle. The length of the secant line ML outside the circle is given as 10 units, while the total length from the external point L to the point **of tangency K is 17 units**. The spatial arrangement is logical, with the circle occupying the central area and the triangle extending outward, creating a clear visual representation of the geometric relationships.

**In detail, the triangle MKL has vertices M, K, and L. Side KM is marked as x, which is also the radius of the circle. Side KL is labeled as 17 units, and** side ML as 10 units**. Points K, M, and L are annotated with bold, black uppercase letters near their respective points, and the variable x appears twice near the two radii and once above the side KM.** The lengths 17 and 10 are written adjacent to sides KL and ML, respectively, in a standard sans-serif font, black in color, and positioned horizontally. The right-angle symbol distinctly marks the perpendicularity of ML to the circle, reinforcing the tangential relationship.

The geometric structure reveals that MK is perpendicular to the tangent line at the point where they meet, a fundamental property of tangents to a circle. The line segment KL acts as a secant line, dividing into two segments: one external (10 units) and one internal (7 units) to the circle. These segments establish the conditions for applying the Power of a Point Theorem, which states that the product of the lengths of the two segments of the secant line equals the square of the length of the tangent from the same external point. **This relationship can be mathematically expressed as:**
$$10 \times 17 = x^2$$
To determine the radius x of the circle, we analyze the right triangle MKL using the Pythagorean theorem. Given that $MK = x, KL = 10$, and $ML = 17$, we can express the relationship as:
$$x^2 + 10^2 = 17^2$$
Simplifying this equation yields:
$$x^2 + 100 = 289$$
$$x^2 = 189$$
$$Therefore, x = \sqrt{189}$$
The image effectively combines visual and textual elements to convey the mathematical concept of the geometry of circles and triangles. It targets students or individuals studying geometry by providing a clear and focused presentation of the problem, eliminating distractions and enhancing understanding and problem-solving efficiency. The logical and systematic spatial arrangement, along with the clear definition of lines and angles, facilitates easy interpretation and application of geometric theorems**. The solution obtained, $x = \sqrt{189}$, is derived purely from the provided geometric relationships and visual information, confirming the utility of the diagram for educational purposes**.

Figure 8: **Comparison between powerful GPT-4.1 and our Capflow in describing a geometry image.** The correctness is marked in green, and the incorrectness is marked in red.

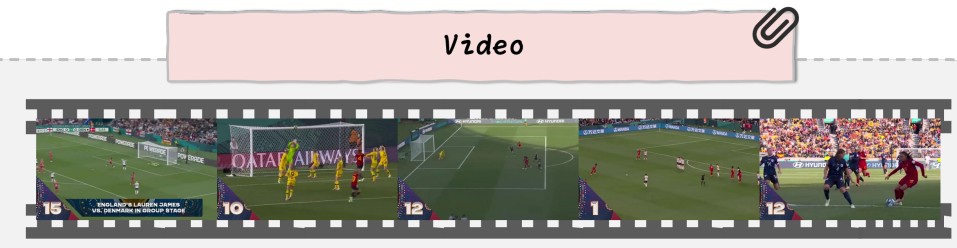

## Video

### GPT-4.1V

Certainly! Here's a detailed description and breakdown of the provided video, which appears to be a countdown or highlight reel of the best goals from the FIFA Women's World Cup 2023:

**General Overview.**

**The video is a dynamic countdown of the top goals scored during the FIFA Women's World Cup 2023, featuring clips from various matches and rounds.** Each segment is dedicated to a specific goal, showing multiple angles and moments from the buildup, execution, and aftermath. There is a graphic overlay in the lower left with the ranking number, and each goal is introduced with an on-screen caption naming the scorer, the opposing team, and the match phase (e.g., group stage, semifinals).

**Detailed Scene-by-Scene Description**

- England's Lauren James vs. Denmark (Group Stage)

    Clips 1–3: The first three images depict England in white and Denmark in red during a group stage match. The action is near Denmark's goal. **The scoreboard shows 0-0 at 5:44. Lauren James is shown scoring a goal from outside the box. One shot shows the ball heading toward the top corner and the Danish goalkeeper diving in vain. Multiple camera angles: wide, behind the goal, and aerial.**

- England's Lauren James vs. China (Group Stage)

    Clips 40–43: **Brazil in yellow, Panama in red, score 2-0 at 47:37.** Zaneratto is involved in a fast attack, finishing with a powerful shot past the goalkeeper. Action highlights her skill and close control.

- Colombia's Linda Caicedo vs. Germany (Group Stage)

    Clips 44–48: **Germany in white, Colombia in purple/yellow, 0-0 at 51:47.** Caicedo dribbles into the box, evades several defenders, and curls a shot into the net. **Celebrations with fans and teammates, and a shot of the ball in the net.**

- Australia's Sam Kerr vs. England (Semifinals)

    Clips 49–53: Australia in yellow, England in blue, **1-0 to England at 62:20.** Kerr makes a solo run, unleashes a long-range shot, and scores a stunning goal. Various angles show the trajectory of the ball and the reaction of fans and players.

- Ireland's Katie McCabe vs. Canada (Group Stage)

    Clips 54–58: Ireland in green, Canada in white/red, **0-0 at 3:34. McCabe scores directly from a corner kick (an "Olympico" goal), with the ball curling into the net.** Celebrations follow, with close-ups of the ball in the net and the reactions.

- Panama's Marta Cox vs. France (Group Stage)

    Clips 59–64: **Panama in red, France in white, 0-0 at 1:05.** Cox takes a long-range free kick, sending the ball into the net for a spectacular goal. **Teammates pile on in celebration as the crowd erupts.**

### CapFlow

This video is a vibrant montage of highlights from the FIFA Women's World Cup 2023, showcasing a series of intense soccer matches and remarkable goal-scoring moments. It features dynamic plays, strategic team efforts, and pivotal individual performances by notable players such as Lauren James, Teresa Abilleira, and Katie McCabe. The video captures the competitive spirit and global excitement of the tournament, with scenes set in vividly depicted stadiums filled with enthusiastic crowds. The visual style combines a clear color scheme and high-quality footage, emphasizing the athleticism and skill of the players against the backdrop of vibrant match settings.

The video begins by setting the stage with a sequence from a match between Panama and France during the Group Stage. The lush green field is teeming with activity as players from both teams position themselves strategically. The focus then narrows onto Marta Cox, Panama's number 10, who initiates an offensive maneuver. Her dribbling and passing skills are on full display, disrupting the French defensive line. As the tension builds, Cox makes a decisive move towards the penalty area, engaging in a critical confrontation with the French goalkeeper. With a deft outmaneuver, she scores a goal that sends the stadium into a frenzy of celebration. This pivotal moment encapsulates the blend of individual brilliance and teamwork that characterizes the tournament.

Expanding further, the video weaves together a collection of similar moments from other matches. Each clip highlights a specific goal, capturing the lead-up actions, the precise execution, and the subsequent celebrations. The players' uniforms and numbers are prominently displayed, facilitating easy identification of key figures like Lauren James, Teresa Abilleira, and Katie McCabe. The ball's trajectory and interaction with players are meticulously captured, adding depth to the portrayal of each critical play. Goalkeepers are often seen in dynamic positions, either diving to intercept or strategically defending the goal. The audience, diverse in their attire and enthusiasm, contributes to the lively ambiance of the event.

The lighting conditions across the clips vary slightly, reflecting different times of day and stadium setups, yet all sequences are uniformly bright, ensuring optimal visibility of the unfolding action. Brand advertisements from sponsors such as Powerade, Dove, and Hyundai adorn the peripheries of the pitch, situating the matches within a broader commercial and global context. The video's editing maintains a seamless flow, with smooth transitions that keep the viewer engaged, enhancing the narrative of the tournament's progression from group stages to quarterfinals and beyond.

The video's thematic essence lies in its celebration of women's football, spotlighting themes of teamwork, individual talent, and the competitive ethos of the sport. The compilation skillfully illustrates the universal appeal and inclusivity of the tournament, featuring a diverse array of teams and players. Camera work plays a crucial role in conveying this message, employing wide-angle shots during build-up phases and close-ups during critical goal moments. This variation in framing allows viewers to appreciate the tactical intricacies and emotional highs of each play.

Analyzing the video's structure and content, it is evident that each segment is carefully crafted to culminate in a climactic goal. For example, one sequence features a free kick that elegantly curves over the defensive wall and sinks into the net. Another showcases a goalkeeper's valiant but unsuccessful dive to stop a shot. These moments underscore the players' technical prowess and the drama inherent in football. The celebratory reactions that follow each goal, ranging from hugs and jumps to joyful sprints, emphasize the significance of these achievements within the context of the tournament.

The video's emphasis on the emotional aspects of the game is further reinforced by the animated responses of the spectators. Their reactions contribute to the overall narrative, reflecting the global appeal and importance of the FIFA Women's World Cup 2023. Through its dynamic visuals, meticulous attention to detail, and thematic depth, the video invites viewers to not only witness but also deeply appreciate the athleticism, passion, and artistry showcased by the female athletes competing on the world stage. The harmonious blend of individual brilliance and collective effort creates a compelling tribute to the spirit of women's football, capturing the heart of this prestigious global event.

Figure 9: **Comparison between powerful GPT-4.1 and our Capflow in describing a sports video.** The correctness is marked in green, and the incorrectness is marked in red.

