# OpenReview forum: "MetaCaptioner: Towards Generalist Visual Captioning with Open-source Suites"
_ICLR.cc/2026/Conference — ICLR 2026 Poster_

### Official Review · Reviewer_V8hc · 2025-10-28

**Soundness:** 3
**Presentation:** 3
**Contribution:** 3
**Rating:** 6
**Confidence:** 3

**Summary:**

The paper proposes a method for generating a small model which is capable of producing high quality captions for real world image data. This requires a complex synthetic data pipeline called CapFlow. CapFlow uses an MLLM domain router along with domain specific agents to generate initial high quality captions for a large amount of data. This is followed by rejection sampling to remove low quality captions. The data is then used to train MetaCaptioner an 8B parameter model which produces high quality captions. The paper compares with several recent open source MLLMs along with the closed GPT4.1 and shows an improvement over the baseline QWEN model at cheaper cost than GPT 4.1.

**Strengths:**

- The paper makes a clear improvement over the baseline QWEN even at 8B
- The automatic captioning pipeline is well motivated and carefully designed
- The method could be useful for data generation tasks of its own

**Weaknesses:**

- Although the paper improved over QWEN the improvement does seem small unless scaling to 72B (Table 2)
- Significant training time (480 + 192 H200 days)

**Questions:**

I think there is a good contribution here but it is a bit hard to understand exactly what the practical impact of the work here is. On the good side I like the design of the CapFlow pipeline from a systems standpoint. The parts all make sense and automating it reduces human expense. I also like what the paper achieves with an 8B model, it's a good goal. That said it wasn't clear to me from the improvements in Table 2 on 8B how much of a difference the model makes in practice. What do the higher scores mean? How do I interpret them? This is coupled with what I would call an extremely expensive training time of 480 + 192 H200 days. That is not cheap, and it may be misleading for Table 2 to show 2c per 100 samples when the actual model training consumed significant energy cost and GPU purchasing cost.

**Specific Questions**
1. How do I understand the results in Table 2 in terms of their practical impact?
2. What is the true cost of MetaCaptioner after accounting for training cost?

---

> ### Author Response · Authors · 2025-11-24
> **Response to Reviewer V8hc**
>
> We feel really honored for your recognition of this work and appreciate your insightful and constructive comments that significantly enhance the quality of our paper. Here are our responses to your concerns and questions, and we hope to gain your further support.
>
> > **Q1: How do I understand the results in Table 2 in terms of their practical impact?**
>
> **A1**: Table 2 presents the ablation of our multi-agent framework, Capflow, in the setting of visual reasoning with LLMs. This setting is designed to evaluate the quality of the caption in an LLM-driven manner. Specifically, the whole evaluation process decouples the visual understanding into two stages: VLLM perception and LLM reasoning. First, the caption model is prompted to generate the caption for images, and then the LLM will use the caption to answer the question directly. Thus, a higher score in this setting can effectively reflect the completeness, accuracy, and  visual consistency of the caption [1][2].  In Table 2, Capflow-72B achieves a comparable performance with the commercial model GPT-4.1 with only 10% cost, which indicates the significant potential of Capflow in the application of large-scale multi-domain data synthesis.
>
> [1] Lu, Yiting, et al. "Omnicaptioner: One captioner to rule them all." Arxiv. 2025.
>
> [2] Qiao, Yuxuan, et al. "Prism: A framework for decoupling and assessing the capabilities of vlms." NeurlPS. 2024.
>
> > **W1: Although the paper improved over QWEN the improvement does seem small unless scaling to 72B (Table 2).**
>
> **A2**: Thanks for pointing this out. In Table 2, we use the OmniCaptioner setup with domain-specific prompts as our baseline, which already achieves fine-grained domain split. In our hierarchical workflow, we initially generate captions using only the natural workflow, which leads to a 1% performance improvement. By further introducing domain routing, we achieve an overall improvement of 1.9%. These gains are quite significant, especially considering that no additional model training is required.
>
> > **W2 & Q2: What is the true cost of MetaCaptioner after accounting for training cost?**
>
> **A3**: We would like to clarify that Table 2 reports the performance and inference cost of the CapFlow pipeline, which is built directly on top of existing open-source model without any additional training. The per-sample cost  are estimated according to the official Qwen2.5-VL-72B API pricing. Concretely, generating 5M captions corresponds to about 480 H200 GPU-days of inference for 72B setting.
>
> In addition, training MetaCaptioner-8B from scratch on these captions requires around 192 H200 GPU-days. This is indeed a one-time training cost. However, once trained, MetaCaptioner-8B is very cheap to deploy: the inference cost is about $0.01 per 100 samples, which remains significantly lower than commercial closed-source models for large-scale captioning. In typical large-scale data synthesis scenarios (millions of captions), this one-time training cost is quickly amortized, and the overall cost remains well below that of repeatedly calling commercial APIs.

---

### Official Review · Reviewer_tFTQ · 2025-10-31

**Soundness:** 3
**Presentation:** 3
**Contribution:** 3
**Rating:** 6
**Confidence:** 4

**Summary:**

The paper introduces CapFlow, a novel multi-agent framework for generalist visual captioning using open-source models, and presents MetaCaptioner, a fine-tuned captioner trained on high-quality data synthesized by CapFlow. The work demonstrates competitive performance with commercial models (e.g., GPT-4.1) at significantly lower cost.

**Strengths:**

1.  MetaCaptioner achieves state-of-the-art results among open-source captioners and approaches GPT-4.1 in quality, especially in reasoning-intensive tasks.
2.	CapFlow demonstrates a substantial reduction in inference and data generation costs compared to commercial models.
3.	A well-motivated hierarchical, domain-specialized multi-agent design that directly tackles domain diversity in visual captioning.

**Weaknesses:**

1. Lack of Direct Human Evaluation. Caption quality is judged by GPT-5; potential model-based evaluation bias is unaddressed.
2. Missing Gemini caption or GPT5 caption comparison. No caption-quality comparison to a closed-source baseline like Gemini, weakening “near-commercial quality” claims.
3. There are hallucinations in reject sampling process. The MLLM (qwen2.5vl-7B) used in the reject-sampling pipeline may exhibit notable hallucinations.
4. It is unclear how specialist and reasoning agents are instantiated and validated (e.g., structural perception, infographic perception, OCR, coder, texture, video perception; general/medical/knowledge/video reasoning). Are these just prompt-specialized variants? How is accuracy ensured relative to visual specialists? Is there cross-agent verification?
5. The workflow appears relatively simple, so tightly binding the approach to a multi-agent framing can feel overstated. A leaner design—e.g., a single controller agent that orchestrates tools and drives self-evolving caption refinement (plan–act–reflect loops)—might achieve similar quality with less complexity and overhead.

**Questions:**

Please see Weakness part.

---

> ### Author Response · Authors · 2025-11-24
> **Response to Reviewer tFTQ (1/2)**
>
> We feel honored for your constructive comments, and appreciate you went through our paper so carefully. Below we try to address your concerns or misunderstandings.
>
> > **W1: Lack of Direct Human Evaluation. Caption quality is judged by GPT-5; potential model-based evaluation bias is unaddressed.**
>
> **A1**: Thanks for pointing this out. We conduct direct human evaluation on Appendix A.4 and Table 9. Results demonstrate strong consistency with the findings in GPT-5 based evaluation result (Table 4). We find that human annotators assigned higher reasoning rigor scores to the fine-grained captions from MetaCaptioner and Capflow compared with OmniCaptioner and GPT-4.1.
>
> > **W2: Missing Gemini caption or GPT5 caption comparison. No caption-quality comparison to a closed-source baseline like Gemini, weakening “near-commercial quality” claims.**
>
> **A2**: We appreciate this constructive comments. We conduct an through comparison between Capflow-72B, Gemini-2.0-Flash, Gemini-2.5-Pro, GPT-4.1, GPT-5 in the setting of visual reasoning with LLMs:
>
> *Table R3: Caption quality comparison between Capflow and other commercial model.*
> | Model             | LLMs         | MMMU | MathVista | AI2D | Overall |
> |-------------------|--------------|:------:|:-----------:|:------:|:---------:|
> | GPT-5             | DS-Qwen-7B   | 58.2 | 64.5      | 74.2 | 65.6    |
> | Gemini-2.5-pro    | DS-Qwen-7B   | 56.8 | 63.8      | 76.2 | 65.6    |
> | GPT-4.1           | DS-Qwen-7B   | 55.7 | 65.0      | 75.5 | 65.4    |
> | Gemini-2.0-Flash  | DS-Qwen-7B   | 56.0 | 56.6      | 74.0 | 62.2    |
> | Capflow-72B       | DS-Qwen-7B   | 55.1 | 62.5      | 74.2 | 64.0    |
>
> As shown in Table R3, Capflow-72B outperforms Gemini-2.0-Flash (64.0 vs. 62.2) and achieves performance comparable to GPT-5 (64.0 vs. 65.6). We observe that Capflow demonstrates a notable advantage in the domain of mathematical images and structure images, e.g., on MathVista, Capflow vs. Gemini-2.0-Flash vs. GPT-5:  62.5 vs. 56.6 vs. 64.5,  on the AI2D benchmark, Capflow even achieve the same performance of GPT-5 (74.2 vs. 74.2), highlighting its strong capability in structured image captioning.
>
> > **W3: There are hallucinations in reject sampling process. The MLLM (qwen2.5vl-7B) used in the reject-sampling pipeline may exhibit notable hallucinations.**
>
> **A3**: Thanks for pointing this out. We have taken concrete steps to minimize hallucinations during the reject sampling process. First, before generating MetaCaptioner-4.1M, we conducted five rounds of human evaluation to carefully refine our prompts. In these rounds, four annotators reviewed both rejected and accepted samples for hallucinations, reasoning quality, visual completeness, and overall human preference. Additionally, we adopted a 3-point scoring system [1] to improve error control and support better decision-making by MLLMs. These efforts have led to a significant reduction in hallucinations at the reject sampling stage. Please refer Appendix A.4 and Table 9 for more human evaluation results.
>
> [1] Fu, Chaoyou, et al. "Video-mme: The first-ever comprehensive evaluation benchmark of multi-modal llms in video analysis." CVPR. 2025.
>
> > **W4: It’s unclear how specialist and reasoning agents are implemented and validated—are they merely prompt-specialized? How is their accuracy compared to visual specialists ensured, and is there cross-verification between agents?**
>
> **A4**: In the design of Capflow, we adopted the state-of-the-art MLLM Qwen2.5-VL-72B and created specific prompt templates for different caption tasks, allowing Qwen to serve in various agent roles. You can find the detailed implementation in https://anonymous.4open.science/r/ICLRRepo-8D68/.
> We argue that Qwen2.5-VL-72B already achieve SOTA multi-modal perception and reasoning ability, e.g., on OmniDocBench [2] (Qwen2.5-VL-72B vs. PPstructure-v3 [3]: 0.214 vs. 0.152), MedBench [4] (Qwen2.5-VL-72B vs. MedGemma [5]: 53.1 vs. 43.2), and MLVU [6] (Qwen2.5-VL-72B vs. Video-XL [7]: 74.6 vs. 64.9). Moreover, since there are very few visual caption models designed for specific domains, we chose Qwen2.5-VL-72B as the base model for our framework.
>
> [2] Ouyang, Linke, et al. "Omnidocbench: Benchmarking diverse pdf document parsing with comprehensive annotations." CVPR. 2025.
>
> [3] Cui, Cheng, et al. "Paddleocr 3.0 technical report." arXiv. 2025.
>
> [4] Ding, Jinru et al. “MedBench v4: A Robust and Scalable Benchmark for Evaluating Chinese Medical Language Models,
> Multimodal Models, and Intelligent Agents.” arxiv. 2025.
>
> [5] Sellergren, Andrew, et al. "Medgemma technical report." ArXiv. 2025.
>
> [6] Zhou, Junjie, et al. "Mlvu: Benchmarking multi-task long video understanding." CVPR. 2025.
>
> [7] Shu, Yan, et al. "Video-xl: Extra-long vision language model for hour-scale video understanding." CVPR. 2025.

---

> ### Author Response · Authors · 2025-11-24
> **Response to Reviewer tFTQ (2/2)**
>
> > **W5: The workflow appears relatively simple, so tightly binding the approach to a multi-agent framing can feel overstated. a simpler design with a single controller agent coordinating tools and iterative caption refinement could offer similar quality with less complexity.**
>
> **A5**: We appreciate this constructive and valuable suggestion. We will incorporate more explanations in the manuscript to better illustrate the design of our framework. The learner-based approach is indeed an interesting and orthogonal direction to our research. However, it currently faces significant challenges, the most critical being the lack of sufficient high-quality trace data for training, which is still unavailable in the community. In contrast, Capflow is a training-free framework that is simpler to implement and does not require additional training data or model fine-tuning, which aligns with our original design intention. However, as you suggestion, we plan to explore the learner paradigm and integrate it with Capflow to develop a more powerful visual captioning model.

---

### Official Review · Reviewer_U7rU · 2025-10-31

**Soundness:** 3
**Presentation:** 3
**Contribution:** 3
**Rating:** 6
**Confidence:** 3

**Summary:**

This paper introduces CapFlow, a multi-agent, hierarchical workflow for generalist visual captioning across nine domains. A domain router selects a domain-specific workflow; functional agents produce evidence; a summary agent aggregates it into a dense caption. Using CapFlow as a data engine, the authors synthesize MetaCaption-4.1M and fine-tune MetaCaptioner-8B. They report caption quality comparable to GPT-4.1 while cutting average inference cost to 10.5%, with further gains when scaling functional agents to 72B.

**Strengths:**

1. Concrete workflows for nine visual domains with explicit agent roles.
2. A multi-agent pipeline used to synthesize 4.1M high-quality captions.
3. Ablations show gains from hierarchical workflows and domain routing.
4. ~89.5% cost reduction vs. GPT-4.1.
5. Implementation details for data and training are provided.

**Weaknesses:**

1. Evaluation leans heavily on LLM-as-a-judge. Most headline claims rely on model-based scoring. This makes the results vulnerable to prompt/model bias. I would expect at least a small but representative human preference study per domain and a few task-grounded metrics to triangulate quality.
2. Router robustness is unclear. The domain router is main contribution, yet there’s no analysis on mixed-domain inputs.

**Questions:**

see weakness

---

> ### Author Response · Authors · 2025-11-24
> **Response to Reviewer U7rU**
>
> Thank you for recognizing our work and providing insightful comments that significantly enhance the quality of our paper. Here are our responses to your concerns and questions, and we hope to gain your further support.
>
> > **Q1: Evaluation leans heavily on LLM-as-a-judge. Most headline claims rely on model-based scoring. This makes the results vulnerable to prompt/model bias. I would expect at least a small but representative human preference study per domain and a few task-grounded metrics to triangulate quality.**
>
> **A1**: Thank you for your constructive comments. We conduct the human evaluation on Appendix A.4 and Table 9. Specifically, we invite 8 human annotators to assess the 250 samples with complex visual scenes. Annotators are asked to follow a rigorous three-point rating protocol for factual accuracy, completeness, reasoning rigor, intent capture, professionalism, and a preference score.  The result demonstrates that our Capflow and MetaCaptioner can achieve a higher score in both reasoning rigor and preference score compared with GPT-4.1 and OmniCaptioner.
>
> As for the task-grounded metrics, we have conducted the corresponding experiments on Table 2 for Capflow and Table 5 for MetaCaptioner. In the settings of visual reasoning with LLMs, we decouple the visual reasoning tasks into two stages: perception and reasoning, which first use VLM models to extract visual information in caption format and then use the corresponding caption as the input of LLMs to respond to questions. A higher score in this setting indicates better quality of caption [1][2].  In this work, we evaluate caption quality in various visual tasks, including natural images (MMMU, MMVet, and MMStar), infographic images (InfoVQA, AI2D, and SEED2Plus), math images (MathVerse, MathVista, and MathVison), and video (MMBench-Video and VideoMME). We have revised the paper, and explict explain the detailed setting in Sec 5.3.
>
> [1] Qiao, Yuxuan, et al. "Prism: A framework for decoupling and assessing the capabilities of vlms." NeurlPS. 2024.
>
> [2] Lu, Yiting, et al. "Omnicaptioner: One captioner to rule them all." Arxiv. 2025.
>
> > **Q2:  Router robustness is unclear. The domain router is the main contribution, yet there’s no analysis on mixed-domain inputs.**
>
> **A2**: Thank you for your valuable insight. We will address this issue through the following two perspectives:
>
> 1.Routing prompt and confidence mechanism. In the design of the routing prompt, we guide the domain router to choose the domain strictly according to the main characteristics of each workflow and introduce a 3-point confidence score mechanism to further verify the result.
>
> 2.Ablation study on routing robustness. We conduct an ablation study to evaluate the robustness of the Domain router. Specifically, four human annotators carefully select 500 images covering 8 major visual domains from the original training data. These images contain complex visual scenes with mixed-domain characteristics (e.g., geometry figures with surrounding text and equations, posters that combine natural scenes with overlaid text, etc.). We perform blind labeling and cross-validation to construct the GT domain labels. After the captioning stage, all four annotators assess both the routing decision and the caption quality for each misrouted sample to further quantify the performance drop (evaluation metrics are consistent with Appendix A.4).
> We use Qwen2.5-VL-7B as the domain router and obtain an overall routing accuracy of 84.8%, while the human average reaches 97.0%. Detailed results for each domain are reported in Table R2:
>
> *Table R2: Domain routing accuracy with Qwen2.5-VL-7B.*
> | Task                    | Accuracy |
> |-------------------------|:----------:|
> | Human Average           | 97.0%    |
> | Average                 | 84.8%    |
> | Natural                 | 85.7%    |
> | Structure & Math        | 87.5%    |
> | Infographic & Document  | 83.3%    |
> | Medical & Bio-Imaging   | 100%     |
> | UI & Interaction        | 77.2%    |
> | Code & Programming      | 83.2%    |
> | Knowledge & Education   | 82.9%    |
> | Synthetic               | 80.0%    |
>
> For the 76 misrouted samples observed in our experiments, we conducted further human evaluation. Despite routing errors, the captions generally maintained a professional standard in most cases: four human annotators rated 64 out of the 76 misrouted samples as acceptable (84.2%). In the final quality filtering stage, 83.3% misrouted samples exhibiting clear hallucinations or errors are accurately excluded.
>
> Although mixed domains can be highly misleading for the model, Capflow is still able to generate professional-level captions by decomposing complex visual tasks through a hierarchical framework, and effectively filters out most samples with hallucinations.

---

### Official Review · Reviewer_5uKz · 2025-10-31

**Soundness:** 3
**Presentation:** 3
**Contribution:** 2
**Rating:** 4
**Confidence:** 4

**Summary:**

This paper introduces CapFlow, a multi-agent workflow for generating visual captions using open-source multimodal large language models (MLLMs). CapFlow employs domain routing and specialized agents to produce multi-domain visual captions, which are filtered via a reject sampling pipeline. The synthesized data is then used to fine-tune MetaCaptioner, a general-purpose captioning model.

**Strengths:**

* The manuscript is clearly written, with a logical flow that makes the overall pipeline easy to follow. The problem statement is well-motivated and practically relevant, making the work engaging and interesting to read.
* If the dataset and models are publicly released as claimed, they would provide a valuable resource for the open-source multimodal learning community, enabling further research on generalist captioning and multimodal alignment.

**Weaknesses:**

* Technical novelty is limited. Although the agent composition and hierarchical workflow are carefully designed, the core components are incremental integrations of existing methods rather than fundamentally new algorithmic contributions. The work introduces no novel learning architecture or objective formulation.

* Although the paper claims that MetaCaptioner achieves performance comparable to GPT-4.1, the presented benchmark results do not fully substantiate this claim.

* Qualitative or diagnostic discussion of where prior models fall short and how the proposed method specifically addresses those limitations is needed.

**Questions:**

* The paper attributes notable performance improvements to the proposed domain routing mechanism. Could the authors provide more analysis or ablations demonstrating its robustness? Specifically, I wonder how sensitive is the system to routing errors or misclassification of domains. Is there a significant drop in caption quality when the routing assigns incorrect workflows?

* While Table 5 shows that captions generated by MetaCaptioner substantially enhance downstream reasoning when used as inputs to external LLMs, Table 6 indicates relatively modest gains in its direct multimodal evaluation. Does this imply that MetaCaptioner mainly improves as a captioning front-end rather than as a genuinely stronger multimodal learner? If so, how does this affect its practical utility in end-to-end multimodal understanding tasks?

---

> ### Author Response · Authors · 2025-11-24
> **Response to Reviewer 5uKz (1/2)**
>
> We thank the reviewer for the kind consideration and constructive comments. We are encouraged that the reviewer recognized our work's clear motivation, strong community contribution, and engaging presentation. Below we try to address your concerns or misunderstandings.
>
> > **W1: The technical novelty is limited; while the agent composition and hierarchical workflow are well designed, the core components are incremental integrations of existing methods rather than fundamentally new algorithms or objectives.**
>
> **A1**: We thank the reviewer for this constructive feedback. The main contribution of CapFlow is a multi-agent caption framework that enhances open-source MLLMs for general-purpose visual description. High-quality visual description data is crucial for MLLMs across a wide range of multimodal understanding tasks. However, large-scale annotation with commercial models such as GPT‑4.1 is prohibitively expensive, while existing open-source models struggle to generate accurate and diverse descriptions for non-natural images, resulting in a substantial performance gap compared to commercial solutions.
>
> To address this limitation, CapFlow decomposes the visual description process into a series of specialized sub-tasks and aggregates evidence from multiple perspectives, thereby improving the descriptive capability of open-source MLLMs. In addition, we introduce a domain routing mechanism that dynamically assigns the most appropriate multi-agent workflow to each visual domain, enabling the framework to adapt to heterogeneous visual inputs and domain-specific requirements.
>
> We note that the multi-agent paradigm has recently been accepted widely in recent visual understanding task accepted by ICML 2025, NeurIPS 2025, and CVPR 2025 (e.g., MetaGPT [1], CapMAS [2], MosAIC [3], SciAgent [4], AgentNet [5]). This trend suggests that multi-agent frameworks are becoming a common design pattern for visual understanding systems, and CapFlow situates itself within this emerging paradigm by targeting general-purpose visual description with open-source MLLMs.
>
> [1] Hong, Sirui, et al. "MetaGPT: Meta programming for a multi-agent collaborative framework." ICLR. 2023.
>
> [2] Lee, Saehyung, et al. "Toward robust hyper-detailed image captioning: A multiagent approach and dual evaluation metrics for factuality and coverage." ICML. 2025.
>
> [3] Bai, Longju, et al. "The power of many: Multi-agent multimodal models for cultural image captioning." ACL. 2025.
>
> [4] Ma, Yubo, et al. "Sciagent: Tool-augmented language models for scientific reasoning." Arxiv. 2024.
>
> [5] Yang, Yingxuan, et al. "Agentnet: Decentralized evolutionary coordination for llm-based multi-agent systems." NeuralPS. 2025.
>
> > **W2: Although the paper claims that MetaCaptioner achieves performance comparable to GPT-4.1, the presented benchmark results do not fully substantiate this claim.**
>
> **A2**: We apologize for this confusion and would like to clarify that it is Capflow, instead of MetaCaptioner-8B, that achieves performance comparable to GPT-4.1. We present the comparison results of MetaCaptioner, Capflow-72B, and GPT-4.1 on visual reasoning with LLMs setting in the following table:
>
> *Table R1: Caption quality comparison between various caption models in visual reasoning with LLMs task.*
> | Caption Model      | LLM                | MMMU | MathVista | AI2D | VideoMME |
> |--------------------|--------------------|-----:|---------:|-----:|---------:|
> | GPT-4.1            | DS-R1-Distill-7B   | 55.7 | 65.0     | 75.5 | 26.8     |
> | Capflow-72B        | DS-R1-Distill-7B   | 55.1 | 62.5     | 74.2 | 27.6     |
> | Qwen2.5-VL-7B      | DS-R1-Distill-7B   | 50.7 | 57.6     | 62.4 | 23.9     |
> | OmniCaptioner-7B   | DS-R1-Distill-7B   | 47.5 | 51.7     | 63.2 | 22.9     |
> | MetaCaptioner-8B   | DS-R1-Distill-7B   | 54.8 | 61.5     | 72.7 | 27.2     |
>
> Although there is still a performance gap between MetaCaptioner-8B and GPT-4.1, MetaCaptioner demonstrates strong captioning ability compared to existing open-source models such as Qwen2.5-VL-7B and OmniCaptioner. We will make this distinction explicit in the revised manuscript.

---

> ### Author Response · Authors · 2025-11-24
> **Response to Reviewer 5uKz (2/2)**
>
> > **W3: Qualitative or diagnostic discussion of where prior models fall short and how the proposed method specifically addresses those limitations is needed.**
>
> **A3**: Thanks for the reviewer’s constructive suggestion.
>
> Although current open-source caption models can produce highly quality captions in the natural domain, they still fall short on knowledge-based images (Table 5). This gap arises because complex images require not only object recognition but also knowledge-based reasoning. For example, describing a geometric image involves not only identifying the main objects, but also precisely measuring the spatial relationships between intersection points using geometric knowledge.
>
> To address this limitation, we decouple the complex visual description task into visual perception, reasoning, and summarization tasks and merge the subcaptions as a professional caption. A domain router will dynamically assign specialized agent workflows to different types of visual input according to their characteristics.
> Compared to previous methods, our approach exhibits better scalability, deployability, and higher performance. As shown in Table 2, our Capflow system based on Qwen2.5-72B achieves captioning performance close to that of the commercial GPT-4.1 model across benchmarks such as MMMU (55.1 vs. 55.7), MathVista (62.5 vs. 65.0), AI2D (74.2 vs. 75.5), and VideoMME (27.6 vs. 26.8). More visualization comparison can be found in the Appendix A.6.
>
> > **Q1:  The paper attributes notable performance improvements to the proposed domain routing mechanism. Could the authors provide more analysis or ablations demonstrating its robustness? Specifically, I wonder how sensitive is the system to routing errors or misclassification of domains. Is there a significant drop in caption quality when the routing assigns incorrect workflows?**
>
> **A4**: We conduct an ablation study to evaluate the robustness of the Domain router. Specifically, four human annotators carefully select 500 images with 8 major visual domains from the original training data. Consequently, blind labeling and cross-validation are performed, and consensus is required for any disagreements. After the caption stage, all four annotators assess both the routing and caption results for each misrouting sample to further evaluate the performance drop. We present the detailed evaluation results for each visual domain in Table R2.
>
> *Table R2: Domain routing accuracy with Qwen2.5-VL-7B.*
> | Task                    | Accuracy |
> |-------------------------|:----------:|
> | Human Average           | 97.0%    |
> | Average                 | 84.8%    |
> | Natural                 | 85.7%    |
> | Structure & Math        | 87.5%    |
> | Infographic & Document  | 83.3%    |
> | Medical & Bio-Imaging   | 100.0%     |
> | UI & Interaction        | 77.2%    |
> | Code & Programming      | 83.2%    |
> | Knowledge & Education   | 82.9%    |
> | Synthetic               | 80.0%    |
>
> As shown in Table R2, Qwen2.5-VL-7B achieved an accuracy of 84.8% in the routing process. For the 76 misrouted samples observed in our experiments, we conducted further human evaluation. Despite routing errors, the captions generally maintained a professional standard in most cases: four human annotators rated 64 out of the 76 misrouted samples as acceptable (84.2%). In the final quality filtering stage, 83.3% misrouted samples exhibiting clear hallucinations or errors are accurately excluded.
>
> We argue that, in most cases, misrouting results only in sub-optimal outputs rather than significant quality degradation. Only a small proportion of misrouted samples adversely affect caption quality, and these can be effectively filtered out during the reject sampling stage. We will include a detailed analysis of routing robustness in the revised paper.
>
> > **Q2: While Table 5 shows that captions generated by MetaCaptioner substantially enhance downstream reasoning when used as inputs to external LLMs, Table 6 indicates relatively modest gains in its direct multimodal evaluation. Does this imply that MetaCaptioner mainly improves as a captioning front-end rather than as a genuinely stronger multimodal learner? If so, how does this affect its practical utility in end-to-end multimodal understanding tasks?**
>
> **A5**: Thanks for your constructive comments. In Table 6, MetaCaptioner-8B achieves an average score of 71.1, which is 2% higher than the latest SOTA MLLM InternVL3.5-8B (69.1). This result demonstrates that simply introducing 4.1M high-quality caption data, under the same training setting as InternVL3.5-8B-Instruct, can significantly enhance the ability of multimodal understanding. In contrast, scaling the parameter size to 30B (InternVL3.5-30B-MoE) yields only a 1.8% improvement. These findings indicate that MetaCaptioner is not only a strong captioning frontend but also gains substantial multimodal understanding from high-quality caption data.

---

### Author Response · Authors · 2025-11-30
**To All reviewers and AC**

We sincerely thank all reviewers for their time, effort, and thoughtful evaluation of our paper. In this work, we proposed a hierarchical, domain-specialized multi-agent caption framework, Capflow, to bridge the performance gap between open-source models and commercial models in a large-scale, multi-domain data synthesis scene.

We summarize the positive feedback and our key revisions and discussions below.

### Summary of Reviewers' Positive Feedback

1. **Strong motivation and clarity**: The manuscript is clearly written with a logical flow, and the problem statement is well-motivated and practically relevant (*Reviewers 5uKz, tFTQ*).

2. **Effective system design**: The hierarchical, domain-specialized multi-agent workflow (CapFlow) is well-designed and concrete, demonstrating clear improvements over baselines (*Reviewers U7rU, tFTQ, V8hc*).

3. **Significant cost efficiency**: CapFlow demonstrates a substantial reduction in large-scale data synthesis costs compared to commercial models like GPT-4.1 (*Reviewers U7rU, tFTQ, V8hc*).

4. **Valuable community contribution**: The proposed dataset and models constitute a valuable resource for the open-source community (*Reviewers 5uKz, V8hc*).

### Summary of Key Revisions and Discussion

1. **Validated domain routing robustness**: We introduced a new test set of 500 high-quality samples across 8 diverse data domains to validate the robustness of domain routing mechanism. Confirmed that CapFlow maintains high performance even under routing errors, demonstrating strong robustness in mix-domain scenarios (*Reviewers 5uKz, U7rU*).

2. **Comparsion with commercial baselines**: We extended our comparative evaluation to include the latest  commercial models, including Gemini-2.0-Flash, Gemini-2.5-Pro, and GPT-5, revealing the significant potential of Capflow in large-scale data synthesis (*Reviewer tFTQ*).

3. **Direct human evaluation**: To address concerns regarding potential bias in LLM-as-a-judge evaluation, we performed direct human evaluation with human annotators (*Appendix A.4*). Confirmed the alignment of different judges, verifying the reliability of our evaluation framework (*Reviewers U7rU, tFTQ*).

4. **Cost and performance concern**: We provided an analysis of training and development costs for MetaCaptioner and Capflow, and explicitly analyzed the performance distinction between CapFlow, MetaCaptioner, and GPT-4.1. This analysis underscores the practical affordability and significant efficiency gains of our approach (*Reviewers 5uKz, V8hc*).

5. **Implementation details**: We discuss the details on agent implementation, hallucination mitigation strategies during reject sampling stage, qualitative analysis with previous approaches, and the experiment settings of visual reasoning with LLMs, ensuring reproducibility and clearer system understanding (*Reviewers 5uKz, tFTQ*).

We believe these revisions have thoroughly addressed the reviewers' concerns and significantly strengthened the contribution and clarity of the paper.

Furthermore, we commit to releasing the complete set of code, models, and data in an open-source community to facilitate reproducibility and future research. We deeply appreciate the reviewers' insightful and constructive feedback, and we kindly hope that you take our revisions and rebuttal into consideration for a possible reassessment.

---

### Meta-Review · Area_Chair_XHyV · 2025-12-24

**Summary:**

The paper introduces a workflow of multiple specialized agents for generating multi-domain visual captions using open-source multimodal large language models (MLLMs), named CapFlow. The main idea behind CapFlow is to use rejection sampling to iteratively refine generated captions. The generated captions are then used to finetune a generalist captioning model, named MetaCaptioner, which is evaluated on multiple benchmarks and shown to achieve performance comparable to commercial models like GPT-4.1.

The paper has been praised for being clearly written and well logically structured. In addition, basing the architecture on open models was indicated as a valuable resource for the open-source multimodal learning community.

The main criticisms raised by reviewers have to do with the limited technical novelty: while the hierarchical workflow is appreciated for being carefully designed, the resulting pipeline is primarily composed of existing methods without introducing novel concepts or techniques. In addition, reviewers noted that the presented benchmarks do not completely hold up the claims of achieving "GPT-4 level" captioning performance.
Finally, reviewers noted that computational costs are quite high for implementing the captioning pipeline and training MetaCaptioner. On one hand, the finetuning dataset examined in the paper is substantial (4.1M samples), implying a considerable inference cost by CapFlow, especially since the method will have to large models like Qwen2-72B to achieve strong results (rebuttal discussion confirmed that generating 5M captions corresponds to about 480 H200 GPU-days of inference for 72B setting).
In addition, training MetaCaptioner-8B on the resulting dataset will also be expensive as argued by reviewers and confirmed by the authors: (192 H200 days in total). The authors qualify these costs by clarifying that these are indeed one-time costs, as the small MetaCaptioner-8B is then very cheap to deploy.

**Reviewer Concerns:**

* Addressed in the rebuttal:
  - Concerns on computational cost have been addressed in the rebuttal with a detailed analysis of training and development costs for MetaCaptioner and Capflow
  - Concerns over the heavy reliance on LLM-as-a-judge metrics have been addressed in the rebuttals by providing human evaluation resulting from the work of 8 human annotators rating 250 samples with complex visual scenes along multiple dimensions (factuality, accuracy, completeness, etc.)
  - Requests for ablation and robustness studies regarding the routing mechanism central in CapFlow have been addressed in the rebuttals with the inclusion of a new test set of 500 high-quality samples across 8 diverse data domains
  - Requests for comparisons against closed-source baselines like Gemini and GPT-5 have countered by the rebuttals by providing empirical evaluations on MMMU, MathVista, AI2D against Gemini-2.0-Flash, Gemini-2.5-Pro, GPT-4.1, GPT-5

* Not addressed in the rebuttal:
  - Limited novelty concerns raised by reviewers have been only partially addressed in the rebuttal
  - Needing to scale to Qwen2-72B to achieve strong results not fully addressed
  - High computational costs for implementing the captioning pipeline and training MetaCaptioner have been acknowledged by the authors in the rebuttal, while specifying that these are one-time costs, as the small MetaCaptioner-8B is then very cheap to deploy

**Reviewer Scores:**

| Reviewer | initial score | predicted final score |
|---:|---:|---:|
| 5uKz | 4 | 6 |
| U7rU | 6 | 6 |
| tFTQ | 6 | 6 |
| V8hc | 6 | 6 |

---

### Decision · Program_Chairs · 2026-01-26

Accept (Poster)